# *Drosophila* epidermal cells are intrinsically mechanosensitive and modulate nociceptive behavioral outputs

Jiro Yoshino[1,2,3†], Sonali S Mali[3,4,5†], Claire R Williams[1,3†], Takeshi Morita[3,6], Chloe E Emerson[3], Christopher J Arp[3], Sophie E Miller[3], Chang Yin[1], Lydia Thé[4], Chikayo Hemmi[2], Mana Motoyoshi[2], Kenichi Ishii[2], Kazuo Emoto[2,7]*, Diana M Bautista[3,4,5,8]*, Jay Z Parrish[1,3]*

[1]Department of Biology, University of Washington, Seattle, United States; [2]Department of Biological Sciences, Graduate School of Science, The University of Tokyo, Tokyo, Japan; [3]Division of Education, Marine Biological Laboratory, Woods Hole, United States; [4]Department of Molecular and Cell Biology, University of California, Berkeley, Berkeley, United States; [5]Helen Wills Neuroscience Institute, University of California, Berkeley, Berkeley, United States; [6]Laboratory of Neurogenetics and Behavior, The Rockefeller University, New York, United States; [7]International Research Center for Neurointelligence (WPI-IRCN), The University of Tokyo, Tokyo, Japan; [8]Howard Hughes Medical Institute, University of California at Berkeley, Berkeley, United States

*For correspondence:
emoto@bs.s.u-tokyo.ac.jp (KE);
dbautista@berkeley.edu (DMB);
jzp2@uw.edu (JZP)

†These authors contributed equally to this work

## eLife Assessment

This is **important** work and provides a significant advance in our understanding of mechanosensation in the epidermis. The evidence presented is **convincing** and, barring a few minor weaknesses, strongly implicates activation of epidermal cells and store-operated calcium entry in the activation of nociceptive neurons innervating that tissue. This work will be of broad interest to neurobiologists, epithelial cell biologists, and mechanobiologists.

**Abstract** Somatosensory neurons (SSNs) that detect and transduce mechanical, thermal, and chemical stimuli densely innervate an animal's skin. However, although epidermal cells provide the first point of contact for sensory stimuli, our understanding of roles that epidermal cells play in SSN function, particularly nociception, remains limited. Here, we show that stimulating *Drosophila* epidermal cells elicits activation of SSNs including nociceptors and triggers a variety of behavior outputs, including avoidance and escape. Further, we find that epidermal cells are intrinsically mechanosensitive and that epidermal mechanically evoked calcium responses require the store-operated calcium channel Orai. Epidermal cell stimulation augments larval responses to acute nociceptive stimuli and promotes prolonged hypersensitivity to subsequent mechanical stimuli. Hence, epidermal cells are key determinants of nociceptive sensitivity and sensitization, acting as primary sensors of noxious stimuli that tune nociceptor output and drive protective behaviors.

## Introduction

The ability to detect tissue-damaging noxious stimuli and mount an escape response is essential for survival. Likewise, prolonged hypersensitivity following injury is an important form of plasticity

that protects an animal from further damage. In *Drosophila*, a single class of identified somatosensory neurons (SSNs), class IV dendrite arborization (C4da) neurons, are necessary and sufficient for nociception; inactivating C4da neurons renders larvae insensitive to noxious stimuli whereas activating these neurons drives nocifensive behavior responses (*Hwang et al., 2007*; *Hu et al., 2017*; *Burgos et al., 2018*). A variety of agents that cause tissue damage including UV irradiation and chemical toxins induce long-lasting allodynia and hyperalgesia (*Babcock et al., 2009*; *Boiko et al., 2017*), but this damage-induced hypersensitivity develops on a timescale of hours. *Drosophila* also display acute hypersensitivity noxious mechanical stimuli (*Hu et al., 2017*). However, the cellular and molecular mechanisms underlying mechanical pain hypersensitivity remain enigmatic.

Recent studies demonstrate that epidermal cells work in concert with SSNs to transduce noxious and innocuous physical stimuli. For example, epidermal Merkel cells are mechanosensory cells that signal to sensory neurons to mediate touch transduction (*Maksimovic et al., 2014*; *Hoffman et al., 2018*). Similarly, keratinocytes are directly activated by noxious thermal and mechanical stimuli and release molecules that modulate nociceptor functions (*Chung et al., 2004*; *Koizumi et al., 2004*; *Moqrich et al., 2005*; *Mandadi et al., 2009*; *Liu et al., 2019*; *Sadler et al., 2020*). Furthermore, epidermal cells in invertebrates and vertebrates ensheathe nociceptors in mesaxon-like structures (*Cauna, 1973*; *Chalfie and Sulston, 1981*; *Han et al., 2012*; *Kim et al., 2012b*; *O'Brien et al., 2012*; *Jiang et al., 2019*), and these sheaths may serve as sites of epidermis-nociceptor signaling (*Yin et al., 2021*). Indeed, epidermal ensheathment is required for normal responses to noxious mechanical stimuli in *Drosophila* (*Jiang et al., 2019*). However, whether epidermal cells are directly activated by noxious stimuli and modulate C4da neuronal activity has not been studied.

Here, we examined the capacity of *Drosophila* epidermal cells to drive nociceptor activation and modulate mechanical nociceptive responses. We found that stimulation of epidermal cells, but no other non-neuronal cell types in the larval body wall, evokes activity in a variety of SSNs and triggers nocifensive behavioral responses. Our in vitro and ex vivo calcium imaging experiments demonstrate that epidermal cells are intrinsically mechanosensitive. Using an unbiased genetic screen, we discovered a role for the store-operated calcium (SOC) channel Orai, and its activator Stim in

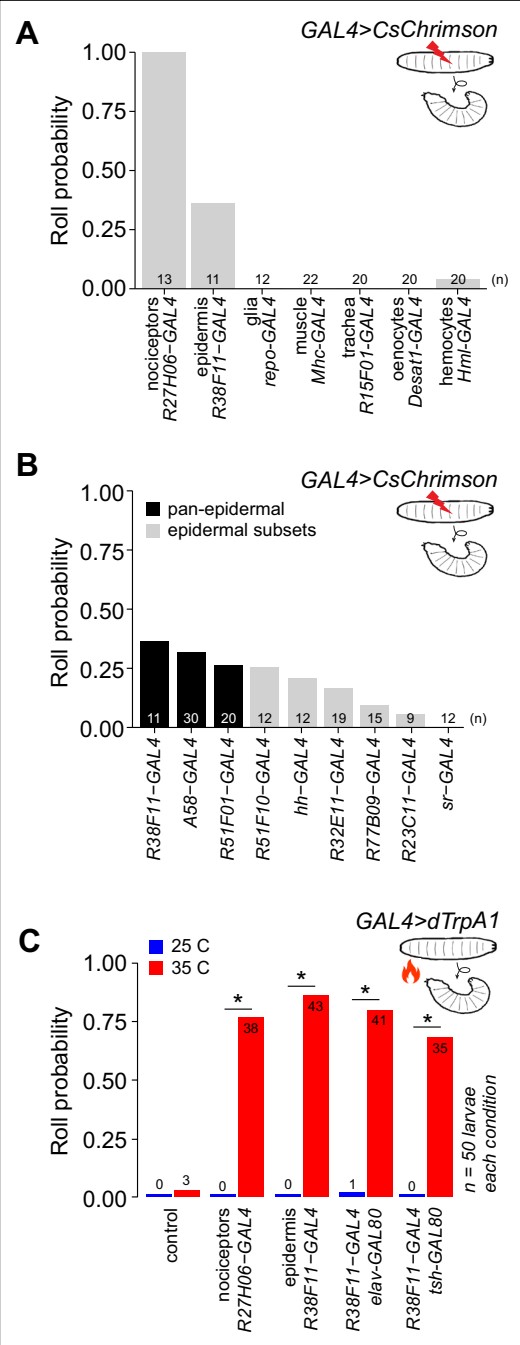

**Figure 1.** Stimulation of epidermal cells elicits nociceptive behaviors. (**A**) Fraction of larvae that exhibited optogenetic-induced rolling (roll probability) using the indicated GAL4 lines to drive *UAS-CsChrimson* expression. All experimental genotypes, except for larvae expressing *UAS-CsChrimson* in class IV dendrite arborization (C4da) neurons, included *elav-GAL80* to suppress neuronal GAL4 activity. Genotypes: *GAL4, UAS-CsChrimson, elav-GAL80/+*. (**B**) Roll probability of larvae following optogenetic stimulation using the indicated GAL4 lines in combination with *elav-GAL80* (or *tsh-GAL80+cha*-GAL80 in the case of *A58-GAL4*) to drive *UAS-CsChrimson* expression

*Figure 1 continued*

in epidermal cells. All epidermal drivers except for *sr-GAL4*, which is expressed in apodemes but no other epidermal cells, elicited rolling responses. Genotypes: *GAL4, UAS-CsChrimson, elav-GAL80/+*. (**C**) Roll probability of larvae following thermogenetic stimulation using the indicated GAL4 line to express the warmth (35°C)-activated *UAS-TrpA1*. The number of rolling larvae (out of 50) is indicated for each group. Genotypes: *GAL4, UAS-TrpA1, GAL80 (as indicated)/+*. Control, *UAS-TrpA1/+*. 50 larvae were tested for each genotype/stimulus combination; the number of rolling larvae is indicated on each bar. Asterisk (*) indicates p<0.05 in this and subsequent figures. Raw data for all figures is provided in *Source data 1* and details of statistical analyses, including tests performed, p-values, and q-values are provided in *Supplementary file 1*.

The online version of this article includes the following figure supplement(s) for figure 1:

**Figure supplement 1.** Related to *Figure 1A*.

**Figure supplement 2.** Related to *Figure 1A*.

**Figure supplement 3.** Related to *Figure 1A and C*.

**Figure supplement 4.** Related to *Figure 1A*.

**Figure supplement 5.** Related to *Figure 1A and C*.

**Figure supplement 6.** Related to *Figure 1B*.

**Figure supplement 7.** Related to *Figure 1B*.

epidermal mechanotransduction and mechanical sensitization. Downstream of Stim/Orai activation, epidermal cells evoke nociceptor activation and mechanical hypersensitivity via epidermal vesicular release. Overall, we demonstrate that *Drosophila* epidermis-neuron signaling mediates both the acute detection of noxious mechanical stimuli and a form of prolonged mechanical hypersensitivity.

## Results
### Stimulation of epidermal cells evokes nocifensive behavior

To identify peripheral non-neuronal cell types that contribute to nociception, we conducted an optogenetic screen for light-evoked nocifensive behavior. First, as a benchmark for comparison we used the light-activated cation channel CsChrimson (*Klapoetke et al., 2014*) to optogenetically activate nociceptive C4da neurons. Consistent with prior reports (*Hwang et al., 2007*; *Hu et al., 2017*), C4da activation triggered nocifensive behaviors including c-bending and rolling in 100% of larvae (*Figure 1A*, *Figure 1—figure supplement 1A*). Next, we selectively expressed CsChrimson using GAL4 drivers in combination with *elav-GAL80*, which effectively silences GAL4 expression in larval sensory neurons (*Figure 1—figure supplement 2*), to target the six principle non-neuronal cell types within the larval body wall: epidermis, trachea, muscle, hemocytes, oenocytes, and glia (*Figure 1—figure supplement 3*, Key resources table). We then monitored light-evoked behavioral outputs associated with stimulation of each cell type. We found that optogenetic stimulation of epidermal cells, like C4da neurons, elicited nocifensive c-bending and/or rolling behaviors in 73% of larvae (*Figure 1A*, *Figure 1—figure supplement 1B*), without significantly altering nociceptor morphogenesis (*Figure 1—figure supplement 4*). In contrast, stimulation of other body wall cell types elicited a variety of non-nociceptive behavior outputs: e.g., muscle stimulation triggered hunching behavior followed by prolonged freezing, whereas glia stimulation reproducibly induced only hunching behavior (*Figure 1—figure supplement 1C–I*; *Zimmermann et al., 2009*). Thus, epidermal cells are the only non-neuronal body wall cell type that triggers robust nocifensive behavioral responses.

To further validate the selective ability of body wall epidermal cells to drive nocifensive behaviors, we examined eight other epidermal drivers in addition to *R38F11-GAL4*, which displays no expression in sensory neurons and limited non-epidermal cell expression overall (*Figure 1—figure supplement 5*). We found that optogenetic stimulation evoked nocifensive behaviors with each of the eight epidermal driver lines we tested: seven of the lines displayed rolling behavior while all eight displayed c-bending (*Figure 1B*, *Figure 1—figure supplement 6*). Although the previously described pan-epidermal *A58-GAL4* driver (*Galko and Krasnow, 2004*) drove robust nocifensive rolling responses (*Figure 1B*, *Figure 1—figure supplement 6*), *A58-GAL4* is expressed broadly in the larval central nervous system (CNS) (*Figure 1—figure supplement 7*) and stochastically expressed in sensory neurons (*Jiang et al., 2014*). In contrast, the remaining seven drivers including *R38F11-GAL4* exhibited limited expression aside from epidermal cells, with no detectable expression in nociceptors, other larval SSNs, or peripheral glia, and highly restricted or undetectable expression in the CNS (*Figure 1—figure supplement 7*). Further underscoring the connection between epidermal stimulation and nocifensive responses, the nocifensive behavioral response with these epidermal drivers correlated with the proportion of epidermal expression (*Figure 1B*).

We next used thermogenetic stimulation with the warmth-activated TRP channel dTRPA1 (*Hamada et al., 2008*) as an independent method of probing nociceptive responses triggered by epidermal cell activation. On its own, the thermal stimulus (35°C) rarely induced rolling behavior in control larvae bearing *UAS-TRPA1* alone. In contrast, we found that >75% of larvae expressing TRPA1 in all nociceptors exhibited rolling behavior in response to a thermal stimulus (*Figure 1C*). Likewise, thermogenetic activation of epidermal cells induced robust rolling responses in >75% of larvae, and addition of GAL80 transgenes (*tsh-GAL80 elav-GAL80*) that silenced the sparse *R38F11-GAL4* VNC expression (*Figure 1—figure supplement 5*) had no effect on the rolling frequency (*Figure 1C*, *Figure 1—figure supplement 3*). Altogether, these results demonstrate that epidermal stimulation evokes nocifensive responses in *Drosophila*. Of note, prior studies demonstrated that sparse thermogenetic activation of nociceptors (<5 cells) yielded no significant increase in nocifensive rolling whereas activation of >10 cells was required to elicit rolling responses in a majority of larvae (*Robertson et al., 2013*). Hence, epidermal stimulation likely engages numerous C4da neurons to elicit these behavioral responses.

In addition to C4da nociceptors, the epidermis is innervated by a variety of other SSNs including mechanosensory C3da and chordotonal (Cho) neurons and proprioceptive C1da neurons. Whereas direct stimulation of C4da nociceptors principally elicited nocifensive behavioral outputs, epidermal stimulation elicited an array of behaviors in addition to nocifensive responses, including freezing and hunching (*Figure 2A and B*, *Figure 2—videos 1 and 2*), behaviors associated with stimulation of C3da and Cho neurons (*Zhang et al., 2013*; *Turner et al., 2016*). These data suggest that epidermal cells may broadly modulate SSN activity in *Drosophila*.

To examine whether different epidermis-evoked behaviors were associated with activation of distinct classes of SSNs, we compared epidermis-evoked and SSN-evoked behaviors. Stimulation of C4da, C3da, and Cho neurons elicited distinct behavioral motifs: only C4da neurons elicited rolling behavior; stimulation of C3da and Cho neurons together elicited hunching, C-bending, and backing; stimulation of Cho neurons alone principally elicited hunching and freezing responses (*Figure 2A–C and F*). In contrast, optogenetic epidermal stimulation elicited all of these behaviors, with nocifensive behaviors (c-bending, rolling) predominating initially, followed by non-nociceptive behaviors (backing, freezing) (*Figure 2D and F*, *Figure 2—figure supplement 1*). We note that neither the behavioral motifs induced by epidermal or SSN stimulation nor the behavioral sequence induced by epidermal stimulation was recapitulated in effector-only controls (*UAS-CsChrimson* ATR+; *Figure 2—figure supplement 1E*), demonstrating that the observed responses were driven by activation of the respective cell types.

We observed three striking differences in behavior evoked by stimulation of epidermal cells versus individual SSNs. First, although rapid, latency to rolling was significantly longer following epidermal stimulation compared to stimulation of C4da (*Figure 2F*). Second, the duration of rolling, bending, and backing responses was significantly longer for epidermis versus SSN stimulation (*Figure 2G*). Third, backing and freezing behaviors persisted beyond the duration of the light stimulus for epidermis but not SSN stimulation (*Figure 2H*). In summary, we find that epidermal stimulation triggers more robust, varied, and prolonged behaviors compared to responses from direct stimulation of discrete SSN subtypes.

## SSNs are activated by epidermal stimulation

We next asked whether epidermal stimulation activates larval SSNs including C4da, C3da, C1da, and Cho neurons. To test this possibility, we developed a semi-intact larval preparation in which we optogenetically stimulated epidermal cells while simultaneously monitoring calcium responses in axon terminals of SSNs (*Figure 3A*). We found that epidermal stimulation triggered rapid and robust calcium transients in nociceptive C4da neurons, responses that were not observed in the absence of ATR or in effector-only controls (*Figure 3B*). Epidermal stimulation likewise evoked calcium transients in mechanosensory C3da and Cho neurons, and in proprioceptive C1da neurons (*Figure 3C–E*, *Figure 3—figure supplement 1*). Hence, epidermal stimulation can broadly modulate activity of larval SSNs.

We next tested the requirement for SSN synaptic transmission in epidermis-evoked behaviors. We stimulated epidermal cells with CsChrimson while blocking SSN neurotransmitter release using tetanus toxin light chain (TnT) (*Sweeney et al., 1995*). We found that inhibiting C4da or C3da+Cho

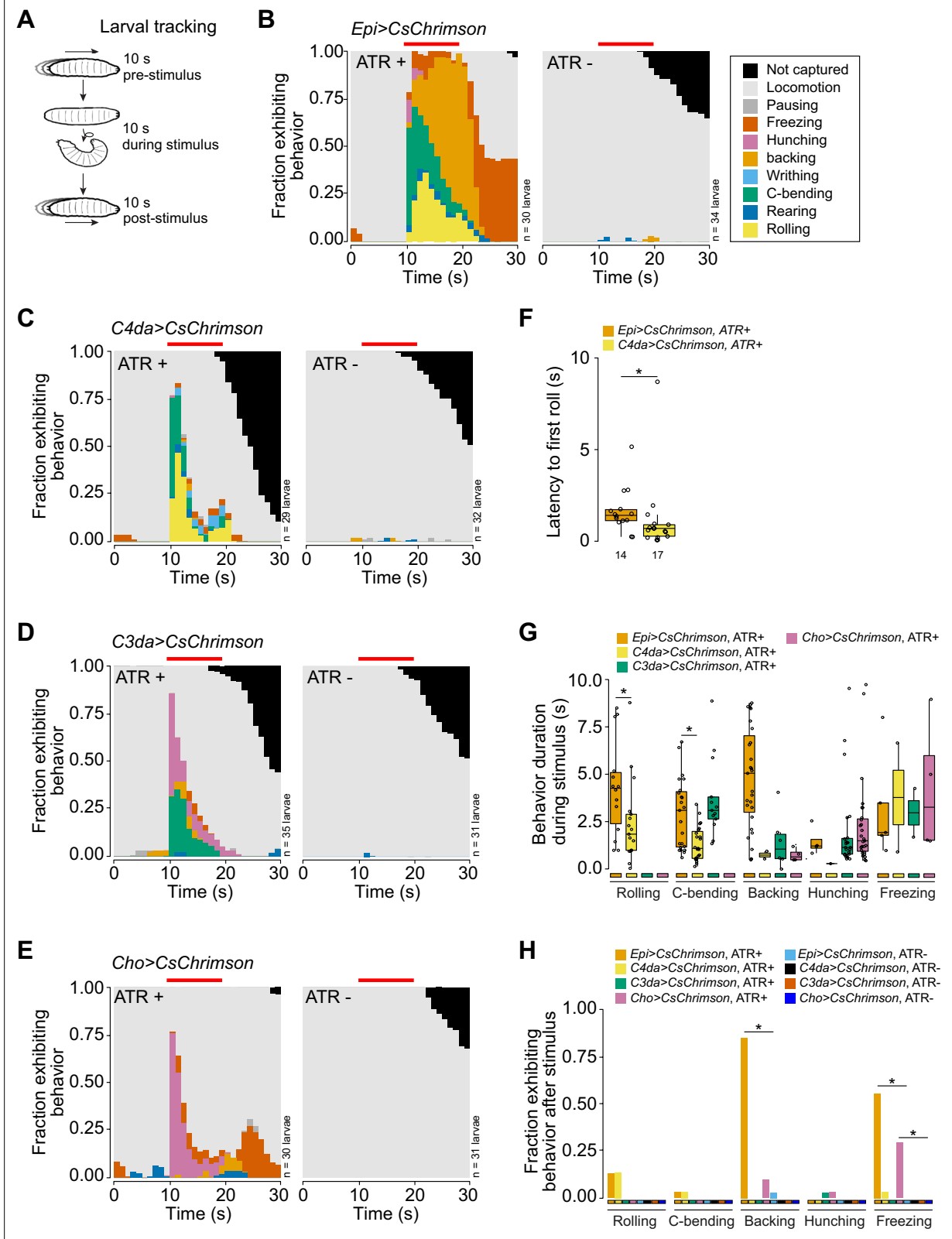

**Figure 2.** Stimulation of epidermal cells evokes multimodal behavioral responses. (**A**) Larval behaviors were scored for 10 s before, during, and after optogenetic stimulation. (**B–E**) Fraction of larvae exhibiting indicated behaviors over time in 1 s bins expressing CsChrimson in (**B**) epidermal cells, (**C**) class IV dendrite arborization (C4da) neurons, (**D**) C3da neurons, and (**E**) Cho neurons in the presence and absence of all-trans retinal (ATR). Red line indicates the presence of light stimulation. (**F**) The latency to the first roll of the larvae that rolled from *Epi>Chrimson* ATR+ and *C4da>Chrimson* ATR+

*Figure 2 continued on next page*

*Figure 2 continued*

treatment groups (n=14, 17, respectively). In this and subsequent box plots, points represent measurements from individual neurons, boxes display the first and third quartiles, hatches mark medians, and whiskers mark 1.5 times the interquartile range. (**G**) The duration of indicated behaviors of the larvae that displayed those behaviors during optogenetic stimulation. (**H**) The fraction of larvae that exhibited indicated behaviors following removal of the light stimulus of all larvae from panels (**B–E**). The number of larvae tested for each genotype/condition is indicated (**B–E**). Genotypes: *GAL4, UAS-CsChrimson/+.*

The online version of this article includes the following video and figure supplement(s) for figure 2:

**Figure supplement 1.** Ethograms and kinetic analysis of behavioral responses to epidermal stimulation.

**Figure 2—video 1.** Behavioral response of representative larva to optogenetic epidermal stimulation.

https://elifesciences.org/articles/95379/figures#fig2video1

**Figure 2—video 2.** Behavioral response of representative larva to optogenetic nociceptor stimulation.

https://elifesciences.org/articles/95379/figures#fig2video2

neurotransmission significantly reduced the frequency and duration of epidermal-evoked rolling and backing behaviors, respectively (*Figure 3F and G*, *Figure 3—figure supplement 2*). These data suggest that C4da and C3da/Cho neurons act downstream of epidermal cells to drive behaviors. We note that TnT expression in C4da neurons did not completely block epidermis-evoked nocifensive behaviors, and this likely reflects both incomplete C4da neuron silencing and epidermal activation of other SSNs that promote nociceptive outputs including C3da neurons, C2da neurons, and Cho neurons (*Ohyama et al., 2015*; *Hu et al., 2017*; *Burgos et al., 2018*). Further, silencing C4da or C3da/Cho neurons while stimulating epidermal cells led to an increase in the non-nocifensive behaviors hunching and freezing (*Figure 3F and G*). These results, along with the observation that rolling behaviors predominate the early behavioral responses to epidermal stimulation (*Figure 2B*), suggest that the nervous system prioritizes nocifensive behavioral outputs following epidermal stimulation. These data support a model in which epidermal cells and SSNs are functionally coupled.

## Epidermal stimulation potentiates nociceptive neurons and behaviors

What is the physiological relevance of this functional coupling of epidermal cells and SSNs? To address this question, we compared calcium responses in C4da neurons to either simultaneous epidermal and C4da stimulation or C4da stimulation alone. Simultaneous stimulation significantly enhanced the magnitude and duration of calcium responses in C4da axons (*Figure 4A–D*, *Figure 4—figure supplement 1*). Based on this prolonged calcium response, we hypothesized that simultaneous epidermis and C4da neuron stimulation would yield enhanced nocifensive behavior output. To test this, we optogenetically stimulated C4da neurons and epidermal cells individually or in combination using low-intensity CsChrimson activation and monitored larval behavior responses. In this stimulation paradigm, simultaneous epidermal cell and C4da neuron stimulation resulted in rolling in 100% of larvae whereas selective stimulation of C4da neurons or epidermal cells induced rolling in only 63% or 18% of larvae, respectively (*Figure 4E and F*). Furthermore, simultaneous stimulation elicited a significantly higher number of rolls among responders than stimulation of nociceptors or epidermal cells alone (26.9 rolls for C4da+Epi, 4.9 for C4da, and 5.3 for Epi stimulation; *Figure 4G and H*). Likewise, simultaneous stimulation significantly reduced the latency to the first roll (*Figure 4I*) and increased the duration of rolling behaviors (*Figure 4J*). We next tested whether this functional coupling extends to mechanical stimuli. We simultaneously presented larvae with a noxious mechanical stimulus and a low intensity optogenetic epidermal stimulus that was insufficient to trigger rolling on its own (0% response rate, n=200). This concurrent epidermal stimulation significantly increased touch-evoked nocifensive responses, yielding a 91% or 49% increase in rolling responses to 20 mN or 50 mN von Frey stimulus, respectively (*Figure 4K*). We next probed the kinetics of this epidermis-induced mechanical sensitization.

   When *Drosophila* larvae are presented with two nociceptive mechanical stimuli in succession, they exhibit enhanced behavioral responses to the second stimulus (*Hu et al., 2017*). We hypothesized that selective epidermal stimulation would sensitize larvae to subsequent nociceptive mechanical stimuli. To test this hypothesis, larvae expressing the warmth-activated calcium-permeable channel dTRPA1 in epidermal cells were presented with a thermal stimulus, 32°C to activate dTRPA1, followed by a 40 mN mechanical stimulus 10 s later (*Figure 4L*). Indeed, we found that dTRPA1-mediated

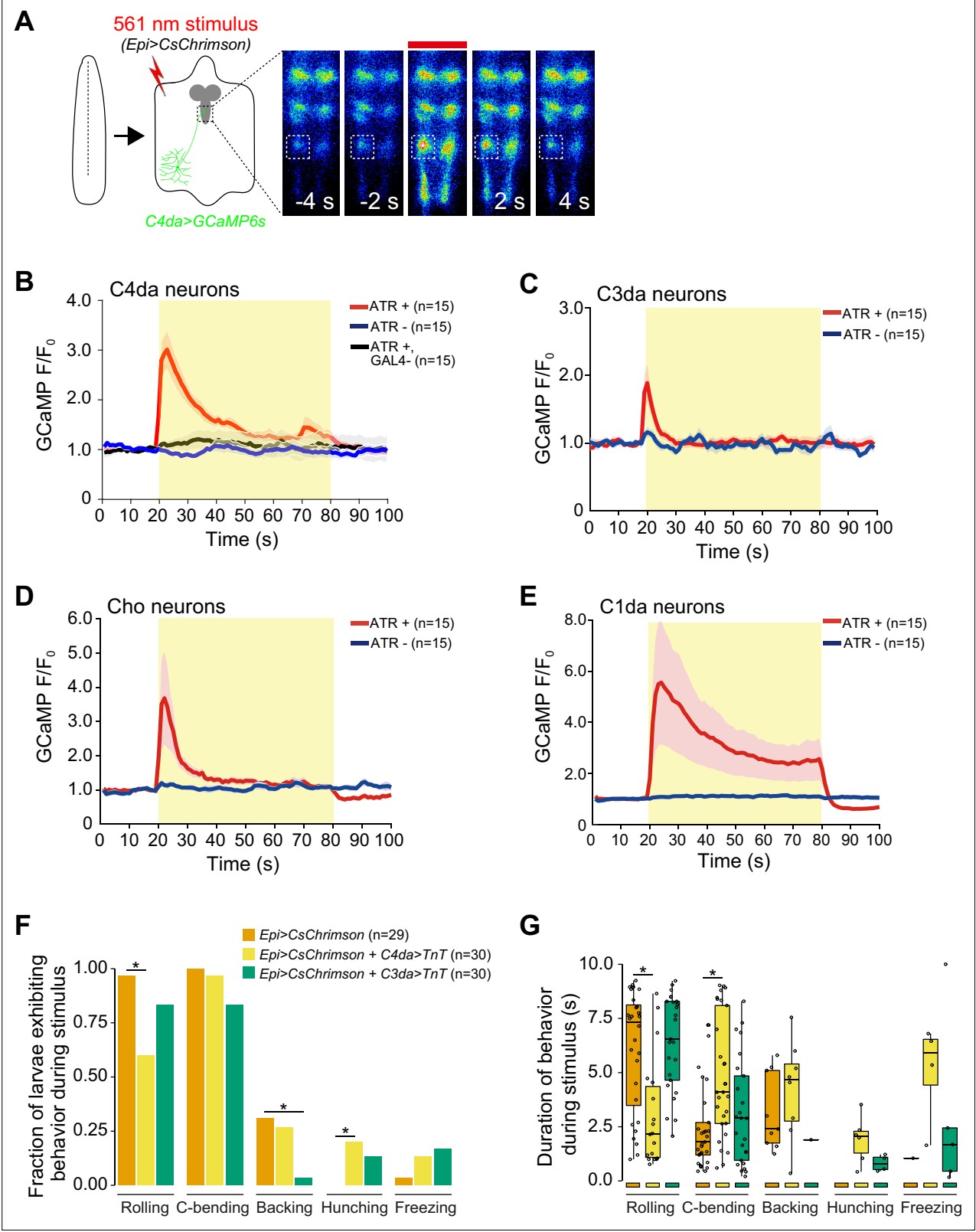

**Figure 3.** Optogenetic epidermal stimulation elicits somatosensory neuron activation. (**A**) Optogenetic activation of CsChrimson-expressing epidermal cells in the body wall triggers calcium transients in the axon terminal of GCaMP6s-expressing nociceptive somatosensory neurons (SSNs). Images show responses from one representative animal. Plots depict mean GCaMP6s fluorescence intensity of the axon terminals of (**B**) class IV dendrite arborization (C4da), (**C**) C3da, (**D**) Cho, and (**E**) C1da neurons following optogenetic activation (light stimulus, yellow box) of epidermal cells over time. Solid lines

*Figure 3 continued on next page*

*Figure 3 continued*

depict mean GCaMP6s fluorescence across replicates (n=15 larval fillet preparations), shading indicates SEM, red traces are *GAL4+* ATR+ , blue traces are *GAL4+* ATR-, black trace is *GAL4-* ATR+. (**F**) The fraction of larvae exhibiting indicated behaviors during optogenetic epidermal stimulation in combination with SSN silencing via tetanus toxin (TnT) expression. We note that although baseline rolling probability is elevated in all genetic backgrounds containing the *AOP-LexA-TnT* insertion, silencing C4da and C3da neurons significantly attenuates responses to epidermal stimulation. (**G**) The duration of the behavioral responses during optogenetic epidermal stimulation with neuronal TnT expression. The number of larvae tested for each genotype/condition is indicated. Genotypes: (**A–B**) *R27H06-LexA* (C4da neurons), *AOP-GCaMP6s, UAS-CsChrimson/+; R38F11-GAL4/+* or *R27H06-LexA* (C4da neurons), *AOP-GCaMP6s, UAS-CsChrimson/+* (GAL4-ATR- effector-only control); (**C**) *AOP-GCaMP6s, UAS-CsChrimson/+; R38F11-GAL4/NompC-LexA* (C3da neurons); (**D**) *UAS-GCaMP6s, AOP-CsChrimson, R61D08-GAL4* (Cho neurons)/*R38F11-LexA*; (**E**) *UAS-GCaMP6s, AOP-CsChrimson, R11F05-GAL4* (C1da neurons)/*R38F11-LexA*; (**F–G**) *R38F11-GAL4, UAS-CsChrimson, AOP-LexA-TnT/+* (*Epi>CsChrimson*); *R38F11-GAL4, UAS-CsChrimson, AOP-LexA-TnT/ppk-LexA* (*Epi>CsChrimson+C4da>TnT*); *R38F11-GAL4, UAS-CsChrimson, AOP-LexA-TnT/NompC-LexA* (*Epi>CsChrimson+C3da>TnT*).

The online version of this article includes the following figure supplement(s) for figure 3:

**Figure supplement 1.** Related to *Figure 3A–D*.

**Figure supplement 2.** Related to *Figure 3E and F*.

epidermal stimulation significantly sensitized larvae to a subsequent mechanical stimulus, increasing the roll probability more than twofold. In contrast, dTRPA1-mediated stimulation of C4da neurons did not induce mechanical sensitization, and we confirmed this result with two independent C4da neuron drivers (*Figure 4L*). Thus, activation of epidermal cells but not C4da nociceptors alone induces prolonged sensitization to noxious mechanical stimuli. We next assessed the duration of sensitization following transient epidermal activation. Thermogenetic epidermal stimulation yielded persistent sensitization that recovered over a timescale of minutes ( $\tau$ =337 s, *Figure 4M and N*). The magnitude and duration of mechanical sensitization by thermogenetic epidermal stimulation was remarkably similar to sensitization evoked by a prior mechanical stimulus (63% roll probability in response to a second stimulus, $\tau$ =334 s, *Figure 4N*, *Figure 4—figure supplement 1B*). Altogether our data support a model whereby epidermal cells are mechanosensitive cells that signal to SSNs to drive acute nocifensive behaviors and prolong mechanical sensitization.

## Epidermal cells are intrinsically mechanosensitive

Prior studies have shown that vertebrate epidermal cells directly respond to mechanical stimuli (*Koizumi et al., 2004*; *Haeberle et al., 2008*; *Tsutsumi et al., 2009*; *Ranade et al., 2014*; *Woo et al., 2014*; *Moehring et al., 2018*). Therefore, we next assessed whether *Drosophila* epidermal cells are intrinsically mechanosensitive. We developed a protocol to acutely dissociate epidermal cells and measure the responses of individual GCaMP6s-expressing epidermal cells to mechanical stimuli (*Figure 5A*). We found that radial stretch elicits calcium responses in epidermal cells in a dose-dependent manner. For example, a low 0.5% stretch activated 18% of cells and a subsequent 1% stretch recruited an additional 10% of stretch-responding cells (*Figure 5B–D*). Overall, 51% of epidermal cells displayed stretch sensitivity (*Figure 5C and D*). We also found that 43% of epidermal cells responded to hypoosmotic challenge and 35% responded to laminar flow; 19% of epidermal cells responded to both hypoosmotic challenge and laminar flow (*Figure 5—figure supplement 1*). Given that dissociated epidermal cells were intrinsically mechanosensitive, we next assessed mechanically evoked responses in a semi-intact body wall preparation (*Figure 5E*). We found that 50% of epidermal cells exhibited a robust calcium transient in response to a 25 µm membrane displacement using a glass probe (*Figure 5E–G*, *Figure 5—figure supplement 1G*). Altogether, these results indicate that *Drosophila* larval epidermal cells are intrinsically mechanosensitive.

## Mechanically evoked epidermal responses rely on SOC entry

Our studies demonstrate that, like vertebrate keratinocytes, *Drosophila* epidermal cells exhibit mechanically evoked calcium transients. What is the mechanism of mechanotransduction in these cells? RNA-seq analysis of acutely dissociated epidermal cells revealed expression of more than 20 cation channels, including the mechanosensitive ion channels Piezo, TMEM63, and TMCO (*Figure 6—figure supplement 1*). We assessed the epidermal requirements of these channels in mechanical nociception using available RNAi transgenes (*Figure 6A*). Our behavioral screen identified one channel, *Orai*, the sole *Drosophila* pore-forming subunit of the $Ca^{2+}$ release-activated $Ca^{2+}$ (CRAC) channel

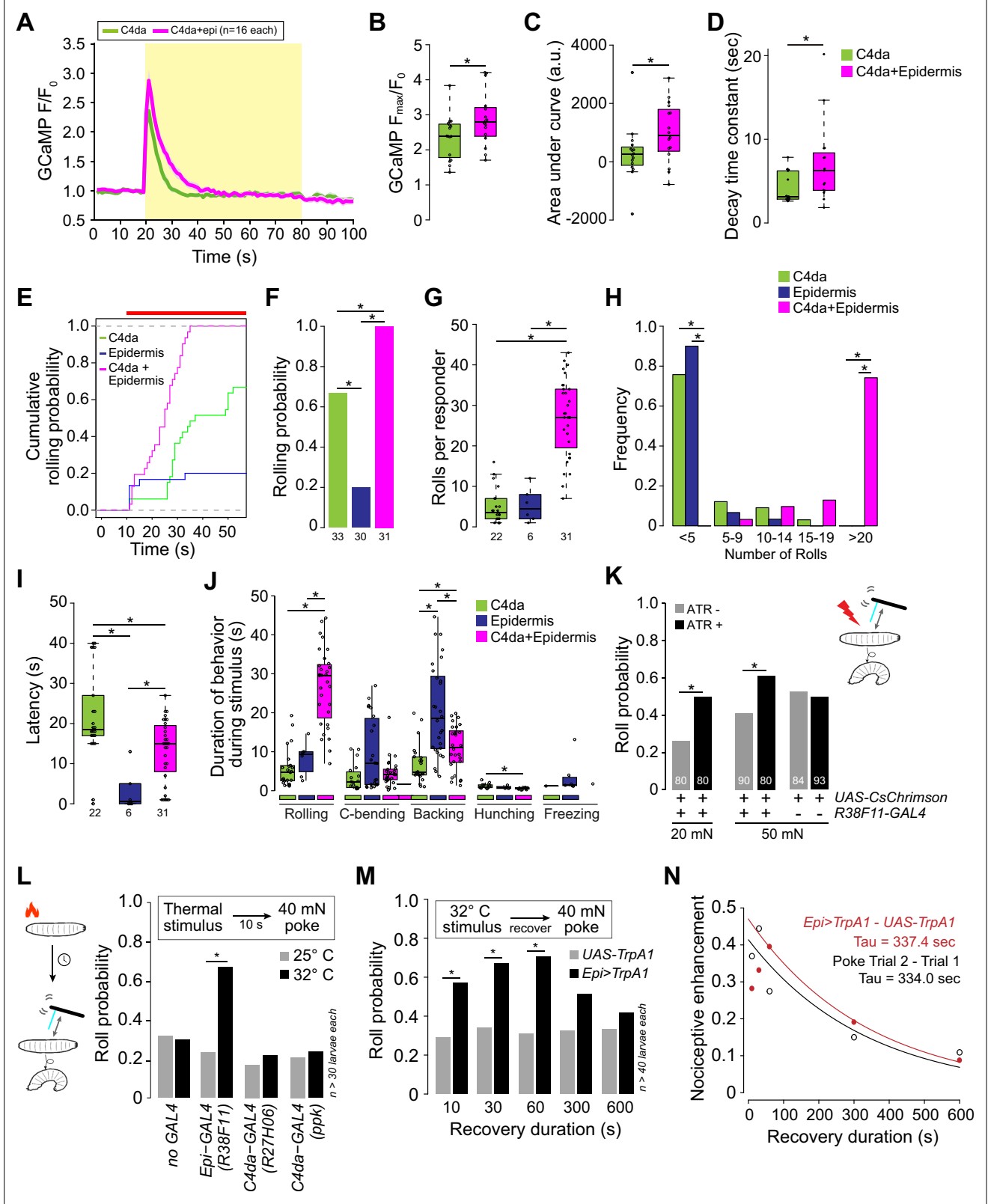

**Figure 4.** Epidermal stimulation augments nociceptive responses. (**A**) Mean GCaMP6s responses ($F/F_0$) in class IV dendrite arborization (C4da) axons during optogenetic stimulation (yellow box) of C4da neurons alone (green) or of C4da neurons and epidermal cells (magenta), shading indicates SEM. (**B**) Simultaneous epidermal stimulation increased the peak calcium response ($F_{max}/F_0$), (**C**) total calcium influx (area under the curve), and (**D**) duration of C4da neuron calcium responses compared to stimulation of C4da neurons alone. Genotypes: *ppk-LexA, AOP-GCAMP6s/+; R27H06-GAL4/UAS-*

*Figure 4 continued on next page*

*Figure 4 continued*

*CsChrimson* (C4da) and *ppk-LexA, AOP-GCAMP6s/+; R27H06-GAL4/R38F11-GAL4, UAS-CsChrimson* (C4da+epi). (**E–J**) Characterization of the behavioral responses to low-intensity optogenetic stimulation of C4da neurons, epidermal cells, or simultaneous C4da neurons and epidermal cells. (**E**) Cumulative and (**F**) total roll probability during optogenetic stimulation (indicated by the red bar). n=33 (*C4da>CsChrimson*), 30 (*Epi>CsChrimson*), and 31 (*C4da+Epi>CsChrimson*) larvae. (**G, H**) Number and frequency distribution of rolls, (**I**) latency to the first roll observed for larvae of the indicated genotypes, and (**J**) the duration of the indicated behaviors during light stimulus. Genotypes: *UAS-CsChrimson/+; R27H06-GAL4/+* (C4da), *UAS-CsChrimson/+; R38F11-GAL4/+* (Epidermis), *UAS-CsChrimson/+; R27H06-GAL4/R38F11-GAL4* (C4da +Epidermis). (**K**) Roll probability of larvae to a 20 mN or 50 mN von Frey mechanical stimulus and epidermal optogenetic activation (a light stimulus, 1.16 µW/mm$^2$ that was insufficient on its own to induce nocifensive rolling). Larvae were reared in the presence or absence of ATR, as indicated. Genotypes: *UAS-CsChrimson/+; R38F11-GAL4/+*. (**L–N**) Prior epidermal but not nociceptor stimulus potentiates mechanical nociceptive responses. (**L**) Roll probability of control larvae (*UAS-TrpA1/+*) or larvae expressing TrpA1 in the epidermis (*Epi-GAL4: R38F11-GAL4*) or C4da neurons (*UAS-TrpA1/+; C4da-GAL4 #1: R27H06-GAL4, UAS-TrpA1/+; C4da-GAL4 #2: ppk-GAL4, UAS-TrpA1/+*), or control larvae (*no GAL4: UAS-TrpA1/+;*) in response to 40 mN mechanical stimulus 10 s following 10 s of a thermal stimulus (25°C or 32°C). To control for effects of genetic background, we confirmed that each of the experimental genotypes exhibited mechanically induced nociceptive sensitization (*Figure 4—figure supplement 1C*). (**M**) Roll probability of control larvae (*UAS-TrpA1/+*) or larvae expressing TrpA1 in the epidermis (*Epi>TrpA1: R38F11-GAL4, UAS-TrpA1/+*) in response to a 40 mN mechanical stimulus delivered at the indicated time interval following a 32°C thermal stimulus. (**N**) Nociceptive enhancement (difference in the roll probability to the first and second stimulus) is plotted against the recovery duration and results were fit to an exponential curve to derive the decay time constant. The red line indicates nociceptive enhancement of a mechanical stimulus by a prior epidermal thermogenetic stimulus; the black line indicates nociceptive enhancement by a prior mechanical stimulus. The number of larvae tested for each genotype/condition is indicated.

The online version of this article includes the following figure supplement(s) for figure 4:

**Figure supplement 1.** Epidermal stimulation induces persistent sensitization of nociceptors.

___

(*Feske et al., 2006*), that blocked mechanically evoked nociceptive sensitization without impacting behavioral responses to the first stimulus (*Figure 6A and B*, *Figure 6—figure supplement 2*) or altering nociceptor morphogenesis (*Figure 1—figure supplement 4*). Interestingly, our screen uncovered an epidermal role for *Task6*, an orthologue of stretch-sensitive two-pore potassium channels (*Fink et al., 1996*), in mechanonociception, as *Task6* RNAi increased nocifensive rolling responses to the initial mechanical stimulus (*Figure 6—figure supplement 2B*). Finally, although our RNAi studies did not reveal an epidermal requirement for other known mechanosensitive cation channels in mechanonociceptive behaviors, it is possible that multiple channels function redundantly, or that RNAi knockdown was incomplete.

To gain insight into mechanically evoked nociceptive sensitization, we focused on probing the role of Orai in epidermal mechanosensory responses. We first asked whether Orai is functional in *Drosophila* epidermal cells. Orai is an SOC channel that is activated by the calcium-sensitive, endoplasmic reticulum (ER) molecule Stim, upon calcium release from ER calcium stores. Thapsigargin (TG) induces calcium release from intracellular stores and thus triggers Stim-dependent activation of Orai channels. Indeed, *Drosophila* epidermal cells displayed TG-induced calcium release from stores in the absence of extracellular calcium, followed by calcium influx upon re-addition of extracellular calcium (*Figure 6C*). Calcium influx was significantly inhibited by the addition of low nanomolar lanthanum, consistent with the high sensitivity of Orai channels to lanthanides (*Figure 6—figure supplement 2C*). This characteristic store-operated calcium entry (SOCE) response was significantly reduced by epidermis-specific *Stim* or *Orai* RNAi knockdown (*Figure 6—figure supplement 2D–F*). Consistent with a key role for SOCE in mechanotransduction, we found that radial stretch in the absence of extracellular calcium induced calcium release from intracellular stores, as well as calcium influx upon re-addition of extracellular calcium. These data show that mechanically evoked responses in epidermal cells involve both ER calcium release and SOCE. While both store release and calcium influx constitute the calcium response to stretch, in 69% of cells, calcium due to store release exceeded that of calcium re-entry (*Figure 6E*). Consistent with this observation, depletion of intracellular stores and inhibition of calcium influx reduced the number of stretch-sensitve cells by 61% (stretch non-responsive cells in WT = 49% vs. store depleted = 80%) and 30% (stretch non-responsive cells in WT = 49% vs. La$^{3+}$ = 64%; *Figure 6F and G*), respectively. Given that Stim and Orai mediate SOCE, we investigated requirements for epidermal Stim and Orai in mechanically evoked calcium responses. RNAi knockdown of either *Stim* or *Orai* significantly reduced the fraction of stretch-responsive epidermal cells (RNAi control = 48%, *Stim* RNAi = 22%, *Orai* RNAi = 24%; *Figure 6H and I*), with *Stim* or *Orai* RNAi preferentially attenuating stretch evoked responses to larger magnitude stretch stimuli. We also found that human keratinocytes display dose-dependent stretch evoked calcium responses, though they

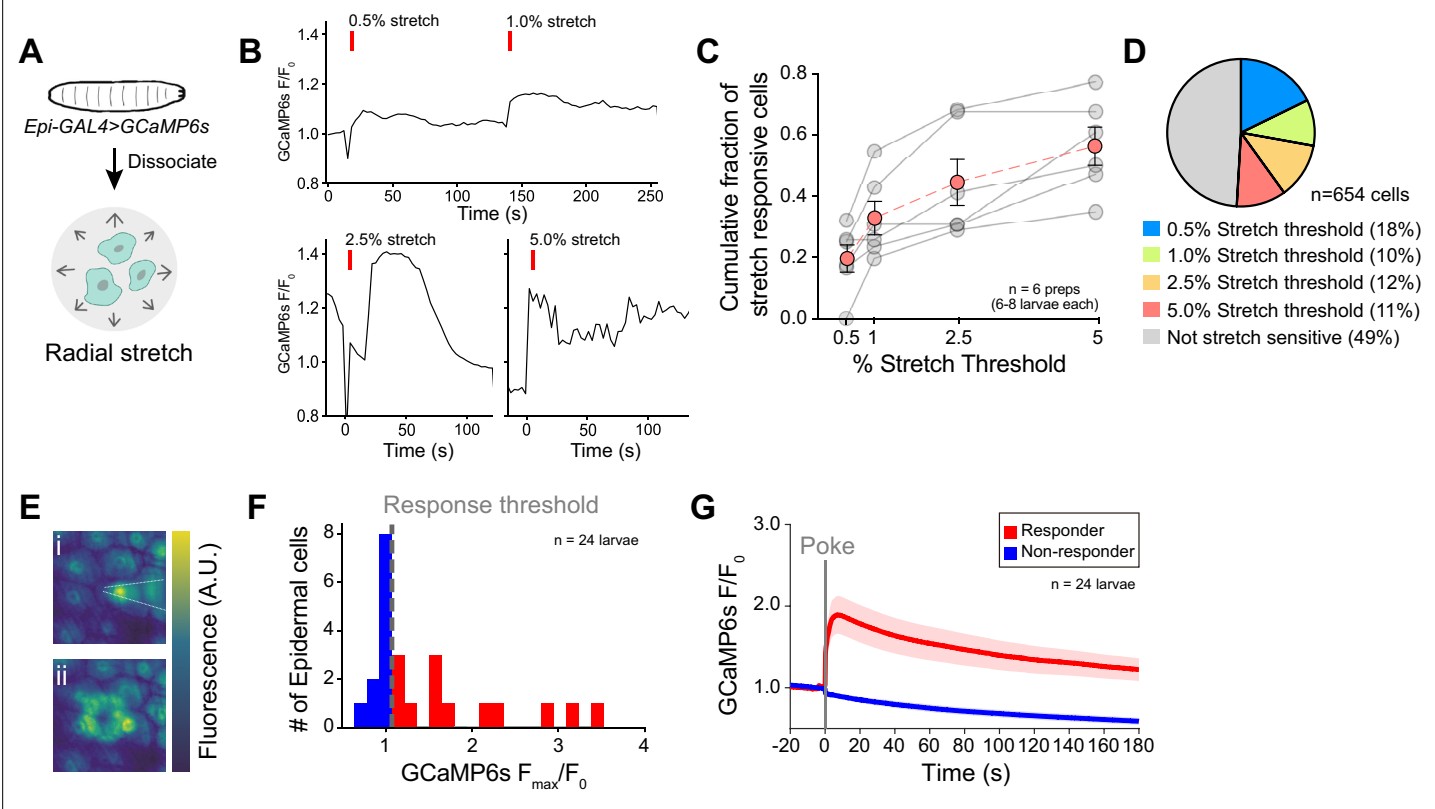

**Figure 5.** Epidermal cells are intrinsically mechanosensitive. (**A**) Schematic of preparation to measure radial stretch evoked calcium responses of dissociated epidermal cells. (**B**) Representative calcium responses of a dissociated epidermal cell to 0.5% and 1% radial stretch (successive stimuli), 2.5% radial stretch, and 5% radial stretch. (**C**) Dose-response curve displaying the fraction of epidermal cells activated by increasing magnitudes of stretch. Red trace displays the mean ± SEM across six independent dissociated cell preparations, obtained from a minimum of six larvae. Gray traces display fraction responding in each dissociated cell preparation replicate. (**D**) Subsets of epidermal cells display varying stretch thresholds, n=6 dissociated cell preparations, for a total of 654 epidermal cells. (**E**) Representative mechanically induced epidermal calcium responses in the larval body wall. Images show GCaMP6s fluorescence intensity 100 ms prior to (i) and 20 s following (ii) a 25 μm membrane displacement (poke). (**F**) Distribution of the peak calcium response ($F_{max}/F_0$) to a 25 μm membrane displacement (poke) of 24 cells from 24 independent larval fillets. Cells were classified as responders (>10% increase in normalized GCaMP6s fluorescence). (**G**) Mean calcium responses ($F/F_0$) of poke responders and non-responders (n=12 cells each). Solid lines depict mean normalized GCaMP6s fluorescence and shading indicates SEM. Sample number is indicated for each genotype/condition. Genotype: *R38F11-GAL4, UAS-GCaMP6s*.

The online version of this article includes the following figure supplement(s) for figure 5:

**Figure supplement 1.** Subsets of epidermal cells display calcium responses to diverse mechanical stimuli.

respond to higher magnitudes of stretch than *Drosophila* epidermal cells (*Figure 6J*). Like *Drosophila* epidermal cells, both ER calcium release and SOCE constitute the mechanically evoked calcium responses in human keratinocytes (*Figure 6K*).

Two hallmarks of Orai channels are steep inward rectification, with larger currents at hyperpolarizing potentials, and highly cooperative Orai activation by Stim (*Hoover and Lewis, 2011*). Since Stim and Orai mediate mechanical responses of epidermal cells in vitro, we predicted that increasing the calcium driving force through Orai activity by either hyperpolarizing *Drosophila* epidermal cells or by activating additional Orai channels via *Stim* overexpression would enhance behavioral responses to mechanical stimuli. Indeed, we found that hyperpolarizing epidermal cells with the light-activated anion channelrhodopsin GtACR1 (*Mohammad et al., 2017*) increased behavioral responses to mechanical stimuli (*Figure 6L*). In addition, overexpressing *Stim* in epidermal cells significantly enhanced nocifensive behavioral responses to mechanical stimuli (*Figure 6M*). Altogether, these results demonstrate that mechanically evoked responses of epidermal cells and the resulting nocifensive behavior outputs require SOCE.

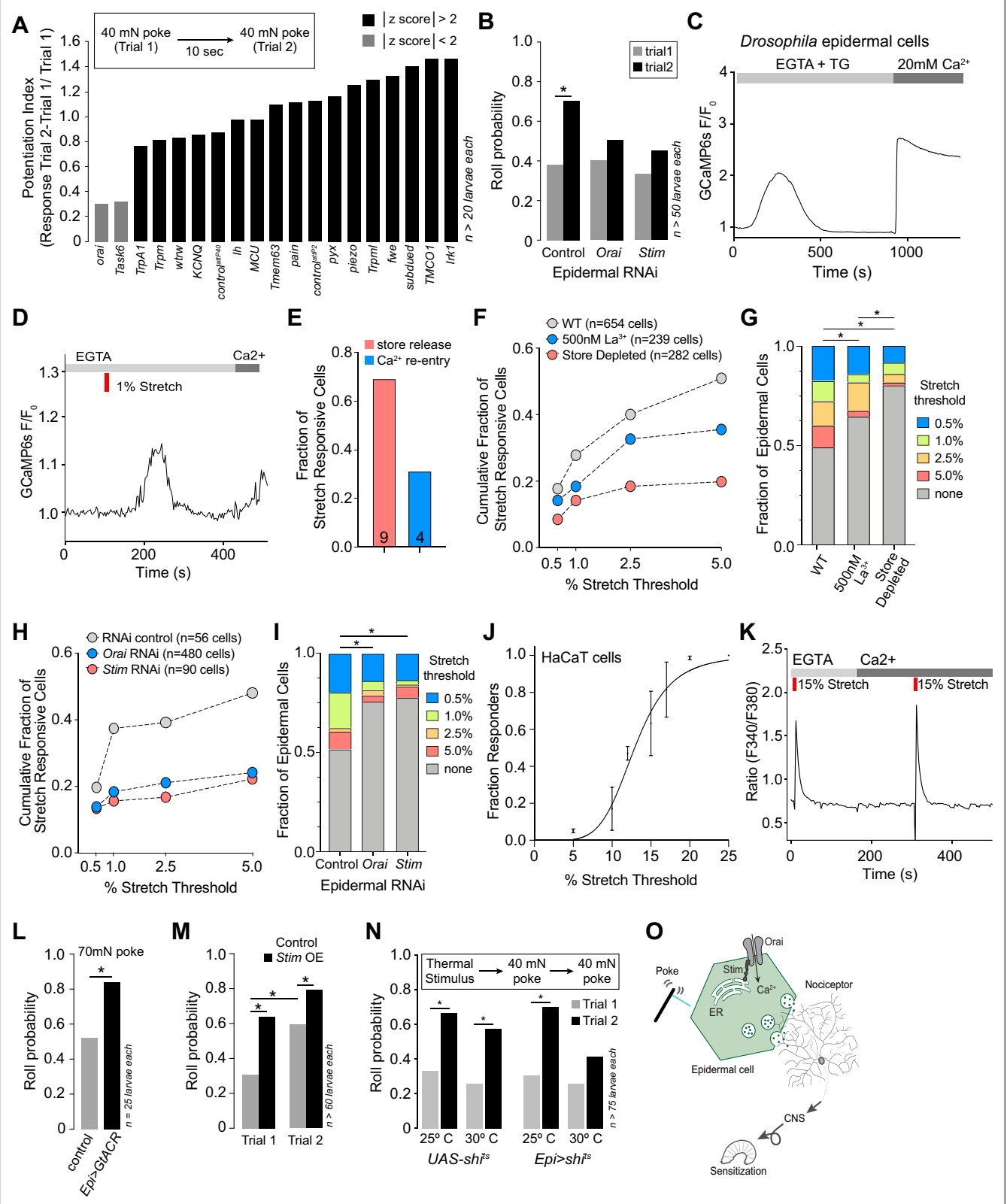

**Figure 6.** $Ca^{2+}$ release-activated $Ca^{2+}$ (CRAC) channels are required for epidermal mechanosensory responses and epidermal nociceptive potentiation. (**A**) RNAi screen for epidermal ion channels required for mechanically induced nociceptive potentiation. Bars depict nociceptive potentiation index (difference in the larval roll probability to the first and second mechanical stimuli divided by roll probability to the first mechanical stimulus). Candidate channels were chosen for further analysis if they had a z-score greater than 2 (absolute value). (**B**) The CRAC channels Orai and Stim are required in

*Figure 6 continued on next page*

*Figure 6 continued*

epidermal cells for mechanically evoked nociceptive potentiation. Roll probability of larvae of the indicated genotypes (control RNAi, *R38F11-GAL4, UAS-RFP-RNAi/+; Stim* RNAi, *R38F11-GAL4, UAS-Stim-RNAi/+; Orai* RNAi, *R38F11-GAL4, UAS-Orai-RNAi/+*) to a 40 mN mechanical stimulus followed by a second 40 mN mechanical stimulus 10 s later. (**C**) *Drosophila* epidermal cells display classical store-operated calcium entry (SOCE). Treatment with the drug thapsigargin (TG) in the absence of extracellular calcium promoted depletion of intracellular calcium stores and calcium influx, following extracellular calcium re-addition. (**D**) Like TG, 1% stretch in the absence of extracellular calcium-induced depletion of intracellular calcium stores and calcium influx, following extracellular calcium re-entry. (**E**) 69% of stretch-responsive cells displayed greater calcium influx during intracellular calcium stores release than during the calcium re-entry phase. (**F–G**) The Orai blocker, lanthanum chloride (500 nM), or the depletion of intracellular stores by TG (1 µM) reduces the fraction of stretch-sensitive epidermal cells. (**H–I**) The fraction of stretch-sensitive epidermal cells is significantly decreased in cells isolated from larvae expressing *Stim* RNAi, or *Orai* RNAi, as compared to control RNAi. (**J**) Stretch stimuli evoke dose-dependent calcium signals in the human keratinocyte HaCaT cell line. (**K**) Representative stretch-evoked SOCE calcium response in HaCaT cells. Stretch induces calcium release from stores in the absence of extracellular calcium and a greater calcium influx in the presence of extracellular calcium. (**L**) Epidermal hyperpolarization enhances mechanical nocifensive responses. Roll probability of larvae expressing GtACR in epidermal cells (*R38F11-GAL4, UAS-GtACR/+*) or control larvae (*R38F11-GAL4/+*) to a single 70 mN mechanical stimulus. (**M**) Epidermal *Stim* overexpression enhances mechanical nocifensive responses. Roll probability of *Stim*-overexpressing larvae (*R38F11-GAL4, UAS-Stim/+*) and control larvae (*R38F11-GAL4/+*) to two successive 40 mN mechanical stimuli delivered 10 s apart. (**N**) Epidermal potentiation of mechanical nociceptive responses requires exocytosis. Roll probability of control larvae (*UAS-shi^{ts}/+*) or larvae expressing temperature-sensitive dominant-negative *shi* in epidermal cells (*R38F11-GAL4, UAS-shi^{ts}/+*) in response to two successive mechanical stimuli that followed 10 min of conditioning at the permissive (25°C) or non-permissive (30°C) temperature. (**O**) Model of epidermal-neuronal signaling. Mechanically evoked Stim/Orai calcium signaling in epidermal cells drives calcium influx and vesicle release that drives nociceptor activation and mechanical sensitization via activation of class IV dendrite arborization (C4da) nociceptors. Sample number is indicated for each genotype/condition.

The online version of this article includes the following figure supplement(s) for figure 6:

**Figure supplement 1.** Related to *Figure 6A*.

**Figure supplement 2.** Mechanically evoked responses of epidermal cells involve SOCE.

**Figure supplement 3.** Epidermal expression of neuropeptide genes and neuropeptide release machinery.

How might mechanically evoked calcium entry in epidermal cells drive nociceptor activation and behavior? Stim/Orai-mediated calcium entry contributes to exocytosis in a variety of cell types, including neurons and immune cells (*Pores-Fernando and Zweifach, 2009*; *Ashmole et al., 2012*; *Maneshi et al., 2020*; *Chanaday et al., 2021*; *Ramesh et al., 2021*). Therefore, we investigated the contribution of epidermal exocytosis in nociceptive sensitization with the temperature-sensitive dynamin mutant *shibire^{ts}* (*shi^{ts}*) to inducibly block vesicle recycling, as this treatment rapidly and potently blocks neurotransmitter release (*Koenig et al., 1983*) and we found that acute epidermal dynamin inactivation using *UAS-shi^{ts}* had no discernable effect on nociceptor morphogenesis (*Figure 1—figure supplement 4*). In this paradigm, larvae expressing *shi^{ts}* in epidermal cells, but not control larvae, exhibited significant attenuation of mechanically evoked nociceptive sensitization following preincubation at the non-permissive temperature (*Figure 6N*). In contrast, both genotypes exhibited comparable responses to a mechanical stimulus at the permissive temperature (25°C) and to the first mechanical stimulus following preincubation at the non-permissive temperature (30°C). Taken together, these results are consistent with a model in which mechanical stimuli induce calcium influx and vesicular release from epidermal cells, which in turn activates nociceptors to induce acute nocifensive behaviors and prolonged sensitization (*Figure 6O*). Although our RNA-seq analysis of epidermal cells did not reveal expression of neurotransmitter biosynthesis genes, epidermal cells express a large repertoire of genes involved in vesicular release as well as several neuropeptide genes, providing an entry point to defining the molecules involved in epidermis-SSN communication (*Figure 6—figure supplement 3*).

## Discussion

In this study, we have shown an essential role for *Drosophila* epidermal cells in escape responses to noxious mechanical stimuli. Activation of epidermal cells acutely activates SSNs to induce an array of behavioral outputs and mechanical sensitization. This epidermal potentiation persists for minutes to promote a prolonged, but reversible, mechanical hypersensitivity that may protect from further insult. This is distinct from previously described forms of neuropathic thermal and mechanical hypersensitivity in *Drosophila* which are induced by tissue damage and chemotherapeutic agents, respectively, emerge on a timescale of hours and are long-lasting (*Babcock et al., 2009*; *Boiko et al., 2017*; *Khuong*

*et al., 2019*). In the mammalian somatosensory system, a variety of inflammatory mediators have been shown to activate TRPA1 in neurons to promote mechanical hypersensitivity (*Bautista et al., 2006*); however, the molecular force transducers that mediate mechanical pain remain unknown. In contrast, in the *Drosophila* somatosensory system, *Ppk1/Ppk26*, *Piezo*, and *Trpa1* are key transducers of mechanonociception (*Zhong et al., 2010*; *Kim et al., 2012a*; *Gorczyca et al., 2014*; *Guo et al., 2014*; *Mauthner et al., 2014*). Prolonged sensitization to noxious mechanical stimuli plays an important protective role in an organism's survival, yet the mechanisms of mechanical sensitization of *Drosophila* nociceptors remain unknown.

We demonstrate a new role for SOC signaling in both *Drosophila* and human epidermal cell mechanotransduction. While short-term sensitization is beneficial to survival, a key hallmark of pathological pain is prolonged and persistent mechanical hypersensitivity; whether deregulation of this mechanism of epidermis-evoked short-term sensitization contributes to pathological pain remains to be determined. Overall, we identified a mechanism that does not impact acute nociception but selectively regulates mechanical sensitization. These findings highlight Stim/Orai signaling as a new avenue for understanding mechanical pain.

This work has opened several new directions for future studies. First, how does radial and osmotic stretch lead to the activation of SOC signaling? Although Orai has not previously been shown to be mechanosensitive, our studies revealed a requirement for Orai and its activator Stim in mechanically evoked calcium flux in *Drosophila* epidermal cells. We also showed that radial stretch of human keratinocytes triggered both calcium release from stores and SOCE; our previous studies showed that Stim and Orai are required for SOCE in human keratinocytes (*Wilson et al., 2013*). These data in combination with other studies showing that mechanical stimulation of human mesenchymal stem cells and mouse enteroendocrine cells (*Knutson et al., 2023*; *Kim et al., 2015*; *Knutson et al., 2023*) also triggers SOCE suggest that Stim/Orai signaling may represent a conserved pathway for mechanotransduction in non-neuronal cells.

Second, how is Stim/Orai function linked to mechanotransduction? Stim/Orai signaling is activated downstream of G-protein-coupled receptors (GPCRs) and receptor tyrosine kinases through phospholipase C. Studies have shown that a number GPCRs are mechanosensitive (*Chachisvilis et al., 2006*; *Grosmaitre et al., 2007*; *y Schnitzler et al., 2008*; *Connelly et al., 2015*; *Xu et al., 2018*). Indeed, this mechanism has been proposed for mechanically evoked enteroendocrine activation in the gut epithelium (*Knutson et al., 2023*), though this has not been studied in epidermal cells. Alternatively, plasma membrane deformation has been shown to induce formation of ER-plasma membrane junctions (*Venturini et al., 2020*; *Aoki et al., 2021*), where Stim and Orai clusters accumulate and interact to drive calcium influx (*Luik et al., 2008*). Finally, a recent paper demonstrated that mechanical stimulation of the ER membrane itself promotes calcium release from ER stores via the opening of calcium-permeable ion channels in the ER membrane (*Song et al., 2024*).

Third, how does mechanically induced signaling in epidermal cells lead to modulation of SSNs? Our data support a model whereby epidermal cells and multiple classes of SSNs are functionally coupled. Epidermal stimulation modulates activity of nociceptive C4da neurons, mechanosensory C3da and Cho neurons, and proprioceptive C1da neurons, and the output of neuronal activity is required for epidermis-evoked behaviors. We demonstrated a requirement for dynamin-dependent vesicle release from epidermal cells in mechanical sensitization, providing a potential link between Stim/Orai signaling in epidermal cells and downstream neuronal activity. However, the mediators that are released by epidermal cells and the signaling molecules in the nociceptors remain unknown. Furthermore, whether different types of SSNs are coupled to epidermal cells by distinct mechanisms remains to be determined. At least in the case of Cho neurons which are wrapped by ensheathing glial cells and scolopale cells, signaling from epidermal cells likely involves at least one additional cell type. Finally, we find that epidermal cells exhibit a dose-dependent response to radial stretch; we therefore anticipate that the output of epidermal cells is likewise dependent on the stimulus intensity. Hence, rather than a fixed threshold beyond which epidermal cells are selectively activated, we hypothesize that increasing stimulus intensities drive increasing signal outputs to neurons.

Epidermal cells ensheathe peripheral arbors of some SSNs, including *Drosophila* nociceptive C4da neurons and, to a lesser extent, mechanosensory C3da neurons (*Jiang et al., 2019*). Hence, epidermal sheaths could facilitate transduction of epidermal signals that modulate nociceptor function. Consistent with this possibility, blocking ensheathment attenuates *Drosophila* larval responses to noxious

mechanical stimuli (*Jiang et al., 2019*) and likewise impairs function of some *Caenorhabditis elegans* mechanosensory neurons (*Chen and Chalfie, 2014*). However, our finding that epidermal stimulation evokes calcium responses from SSNs that are not ensheathed by epidermal cells (C1da, Cho neurons) argues that epidermal sheaths are unlikely to play an essential function in epidermis-SSN functional coupling. Instead, ensheathment may facilitate nociceptor activation by increasing the efficiency of vesicular exchange or, alternatively, may modulate nociceptor activity through enhanced ionic coupling to epidermal cells.

Which epidermal-derived molecules might modulate neuronal activity? There are several mechanisms by which mammalian epidermal cells activate SSNs. Vesicular release of norepinephrine from mouse epidermal Merkel cells is required for sustained touch-evoked firing of mechanosensory neurons (*Hoffman et al., 2018*). Additionally, mechanical stimuli trigger ATP release from mouse keratinocytes that activates nociceptors via purinergic (P2X4) receptors (*Koizumi et al., 2004*; *Tsutsumi et al., 2009*; *Moehring et al., 2018*). Finally, Stim/Orai-dependent SOCE mediates the release of the cytokine thymic stromal lymphopoietin from epidermal keratinocytes that directly activates a subset of TRPA1-expressing SSNs to induce itch (*Wilson et al., 2013*). Similar to these mammalian models, UV damage has been shown to induce the release of the cytokine Eiger to promote *Drosophila* nociceptor sensitization (*Babcock et al., 2009*), though this occurred on a slower timescale than the epidermal-evoked mechanical sensitization we describe here (8 hr vs. ~10 s, *Figure 4L*). Likewise, epidermal platelet-derived growth factor (PDGF) ligands regulate mechanonociceptive responses in *Drosophila* (*Lopez-Bellido et al., 2019*) and intrathecal delivery of PDGF or the closely related growth factor EGFR yields mechanical hypersensitivity in rats (*Masuda et al., 2009*; *Puig et al., 2020*), but it remains to be determined whether growth factor signaling can yield rapid sensitization. Hence, future studies will address which neurotransmitters, neuropeptides, or inflammatory mediators underlie epidermal cell-mediated mechanical sensitization.

Our data support a model whereby epidermal cells and multiple classes of SSNs are functionally coupled. Future studies will address which neurotransmitters, neuropeptides, or inflammatory mediators underlie epidermal cell-mediated mechanical sensitization. An additional key next step is understanding whether the neuronal plasticity underlying mechanical sensitization results from the direct modulation of mechanosensitive channels or rapid insertion of new mechanosensitive channels into the plasma membrane, or from changes in the signaling pathways or channels that regulate neuronal excitability. Overall, we performed an unbiased genetic screen that for the first time establishes a key role for mechanically evoked Stim/Orai calcium signaling in epidermal cells that drive nociceptor modulation and mechanical hypersensitivity.

## Materials and methods
### Materials availability and community standards
Raw sequencing reads and gene expression estimates are available in the NCBI Sequence Read Archive (SRA) and in the Gene Expression Omnibus (GEO) under accession number GSE284380. Raw data used for analyses in this study is presented in the supplementary materials as Source Data and details of statistical analyses are presented in *Supplementary file 1*. ICMJE guidelines were used to define authorship roles and the ARRIVE essential 10 guidelines were used for the reporting of our in vivo studies.

### *Drosophila* strains
Flies were maintained on standard cornmeal-molasses-agar media and reared at 25°C under 12 hr alternating light-dark cycles. For all experiments involving optogenetic manipulations, larvae were raised in the constant dark at 25°C on Nutri-Fly Instant Food (Genesee Scientific #66-117), supplemented with 1 mM all-trans retinal (ATR; Sigma #R2500). A complete list of alleles used in this study is provided in the Key resources table. Experimental genotypes are listed in figure legends.

### Cell lines
A human keratinocyte cell line (HaCaT) was used in this study. HaCaT cells were obtained from Cytion (Sioux Falls, SD, USA), who performed STR authentication and mycoplasma-free certification.

## Behavior analysis

### Optogenetic behavior screen

Individual larvae were rinsed in ddH$_2$O, transferred to an agarose substrate (1% agarose, 100 mm dish) in a darkened arena, and habituated for 30 s. Larvae were stimulated with a top-mounted 488 nM LED illuminator (PE-300, CoolLED) and images were captured with an sCMOS camera (Orca Flash 3.0, Hamamatsu) at frame acquisition rate of 20 fps and behaviors were scored before, during, and after optogenetic stimulation.

### High-resolution video tracking of optogenetic-gated larval behavior

Following 5 min of light deprivation including 15 s of habituation in the behavioral arena, larvae were tracked before, during, and after optical stimulus (10 s each, 30 s total) (*Figure 2A*). For these studies we modified our stimulation paradigm in two key ways: to avoid potential contributions of nociceptor light-evoked responses (*Xiang et al., 2010*), we stimulated larvae using yellow-shifted light; and to facilitate kinetic analysis of behavior outputs, we used an automated shutter. Larvae were stimulated with a top-mounted 585 nm LED illuminator (SPECTRA X, Lumencor) equipped with a filter (FF01 585/40-25, Semrock), and images were captured with an sCMOS camera (Zyla4.2, Andor) at a frame rate of 20 Hz. Larvae were constantly illuminated with an infrared (940 nm) light source (LDR2-132IR2-940-LA, CSS) for visualization. Larvae were fed (ATR+) or vehicle alone (ATR-) as indicated. Illumination intensities for optogenetic behavior studies were: 300 µW/mm$^2$ for *Figure 1—figure supplement 6*, *Figure 2B–E*, *Figure 2—figure supplement 1*, *Figure 3E and F*, *Figure 3—figure supplement 2A–C*; 25 µW/mm$^2$ for *Figure 4E–J*; 1.16 µW/mm$^2$ for *Figure 4K*. Annotated videos showing responses of representative larvae to optogenetic epidermal and nociceptor stimulation are provided in *Figure 2—videos 1 and 2*.

### Thermogenetic behavior assays

Larvae for thermogenetic assays were reared at room temperature (20°C) to limit TRPA1 activation during development. Third-instar larvae were isolated from their food, washed in distilled water, and recovered to damp agar plates for several minutes, and transferred individually to a Peltier plate held at 25°C or 35°C. Behavior responses were recorded under infrared light with a computer-controlled GigE camera (FLIR) at an acquisition rate of 20 fps for 20 s. Responses were analyzed post hoc blind to genotype and were plotted as the proportion of larvae that exhibited at least one complete nocifensive roll during stimulus application.

### Mechanonociception assays

Third-instar larvae were isolated from their food, washed in distilled water, and placed on a scored 35 mm Petri dish with a thin film of water such that larvae stayed moist but did not float. Larvae were stimulated dorsally between segments A4 and A7 with calibrated von Frey filaments that delivered the indicated force upon buckling, and nocifensive rolling responses were scored during the 10 s following stimulus removal. For assays involving multiple stimuli, larvae were stimulated individually, allowed to freely locomote in the arena for up to 1 min (for longer recoveries larvae were recovered onto 2% agar to prevent desiccation), and subsequently presented with the second stimulus. For assays involving thermal and mechanical stimuli, larvae were individually transferred to a pre-warmed Peltier plate containing a thin layer of water, incubated for the indicated time, and transferred to the behavior arena (or a 2% agar plate for recoveries>1 min) with a paint brush for subsequent mechanical stimulation. For assays involving optical and mechanical stimuli, larvae were raised in constant dark at 25°C on food supplemented with 1 mM ATR (detailed above), transferred to the behavior arena with 25 µW/mm$^2$ broad-spectrum illumination, and assayed for responses to mechanical stimuli. All assays were conducted in ambient light except for experiments with GtACR (*Figure 6L*), which were conducted under 500–700 nm LED illumination (CoolLED PE-300, green). Our illumination setup for these experiments provided limited working distance, therefore larvae were restrained with forceps and given only a single stimulus.

### Video annotations

Videos of individual larvae responding to light stimuli were scored on a frame-by-frame basis using the annotation software BORIS (*Friard and Gamba, 2016*). Behaviors scored, along with descriptions of the criteria for each behavior, are detailed in *Supplementary file 2*. Video analysts were blind to the genotype and treatment during scoring. Scoring on a training set was compared across all analysts to calibrate, and any behaviors for which the primary analyst was uncertain were reviewed by an additional analyst. Additionally, 10% of videos were scored independently by two analysts and there was at least 80% concordance in behaviors annotated in these comparisons.

## Microscopy

### Calcium imaging: ventral nerve cords

Third-instar larvae were dissected along the dorsal midline and pinned on a sylgard-coated dish (Sylgard 184, Dow Corning). The internal organs except for neural tissues were removed. Larvae were bathed in HL3.1 (*Feng et al., 2004*) and modified to remove calcium (*Supplementary file 3*) to minimize larval movement. The ventral nerve cord was imaged using an Olympus BX51WI microscope, equipped with a spinning-disk confocal unit Yokogawa CSU10 (Yokogawa) and an EM-CCD digital camera (Evolve, Photometrics). For activation of epidermal cells with the light-gated CsChrimson, red light was delivered by a pE-300 (CoolLED) equipped with a filter (ET645/30×, Chroma) at a light intensity of 30 μW/mm$^2$. Obtained images were analyzed using Metamorph and ImageJ (*Schneider et al., 2012*). Baseline fluorescence was calculated as the mean fluorescence intensity of an ROI over the 10 frames prior to light stimulus delivery. The trapezoidal method was used to calculate area under the curve, utilizing the trapz function of MATLAB. Data points from the onset of stimulation to the end of stimulation were used for the calculation.

### Calcium imaging: fillet preparations

Third-instar larvae were dissected along the ventral midline and pinned on sylgard (Dow Corning) dishes with the internal surface facing toward the microscope. All internal organs, including the CNS, were removed. Larvae were bathed in calcium-containing HL3.1 (*Supplementary file 3*) except where indicated and images of the dorsal midline between abdominal segments A2 and A4 were captured with a Zeiss Axio Zoom V16 microscope. Captured images were analyzed using ç. Mechanical stimulus: fillets were poked with a tapered borosilicate capillary with a rounded tip, using a micromanipulator to induce a deflection of 25 μm. The decay time constant was calculated by fitting the data points from the peak response to the end of the experiment into an exponential curve f(x)=a*exp(b*x) using MATLAB with R$^2$>0.9 used as a threshold for reliable fitting.

### Calcium imaging: dissociated epidermal cells

Six to eight larval fillets were dissociated in 400 μL of 50% saline (modified Ringer's recipe)/50% Schneider's media with 200 U/mL collagenase type I (Fisher 17-100-017), with mixing at 1000 RPM at 33°C for 16 min, with trituration every 8 min. Undigested fillets were removed and the remaining suspension was spun at 500×*g* for 3 min, followed by aspiration of the supernatant down to a 10 μL cell suspension. Cells were resuspended in 30 μL fresh PBS/Schneider's solution and plated onto poly-D-lysine (1 mg/mL, Sigma P7886) coated No. 1 coverslips, with 10 μL cell solution per coverslip. Cells were cultured at least 30 min and up to 2 hr at 25°C prior to imaging. Cells were imaged using a ×10 objective at a frame rate of 0.33 Hz. Solutions are indicated in figure legends (see *Supplementary file 3* for recipes). Obtained images were analyzed using MetaFluor and Python, and baseline fluorescence was calculated as the mean fluorescence intensity of an ROI over 5 frames prior to stimulus delivery. For stretch stimulation, circular membranes were cut with an arch punch from sheets of glossy silicone of 0.01–0.02 inch thickness (Specialty Manufacturing, Inc) and coated with 1 mg/mL poly-D-lysine for 1 hr before plating cells. Membranes were mounted onto the StageFlexer system and vacuum pressure was applied through the FX-3000 system (Flexcell). Calibrations were performed using fluorescent beads attached to the membranes, and images were taken before and during a static stretch. To stimulate cells, a 2 s square wave of vacuum pressure was applied. Cells were imaged with an Olympus BX61WI upright microscope. For SOCE measurements and osmotic stimulation, cells were imaged using a Zeiss Observer inverted microscope and solutions were perfused using the

Automate Scientific ValveLink 8.2 perfusion system. At the end of each imaging session, 1 µM iono-mycin was perfused and only cells that showed a calcium response, as defined by a 10% increase from baseline fluorescence, were used in analysis. Flow, osmotic and radial stretch responders were defined by a 5% increase from baseline fluorescence.

## Calcium imaging: human keratinocytes

Immortalized human keratinocytes (HaCaT) cells (Cytion) were plated on silicone membranes 1 day prior to stretch experiments. Prior to the radial stretch experiments, cells were loaded with 1 µm Fura-2AM supplemented with 0.01% Pluronic F-127 (wt/vol, Life Technologies) in a physiological Ring-er's solution containing the following (in mm): 140 NaCl, 5 KCl, 10 HEPES, 2 $CaCl_2$, 2 $MgCl_2$, and 10 D-(+)-glucose, pH 7.4. Acquired images were displayed as the ratio of 340 nm/380 nm. Cells that had a response 10 standard deviations above baseline to ionomycin were included in the analysis and stretch responses were defined by a 15% increase in Fura-2 340/380 ratio.

## Confocal microscopy

For peripheral imaging of cellular morphology, live single larvae were mounted in 90% glycerol under a coverslip and imaged on a Leica SP5 confocal microscope using a ×40 1.25 NA lens. To image the larval CNS, larvae were dissected on sylgard plates, briefly fixed in 4% paraformaldehyde in PBS for 15 min at room temperature, washed 3×5 min in PBS, and mounted for imaging.

## RNA-seq analysis of epidermal cells

### RNA isolation for RNA-seq

Larvae with cytoplasmic GFP expressed in different epidermal subsets were microdissected and dissociated in collagenase type I (Fisher 17-100-017) into single-cell suspensions, largely as previously described (**Williams et al., 2016**), with the addition of 1% BSA to the dissociation mix. After dissociation, cells were transferred to a new 35 mm Petri dish with 1 mL 50% Schneider's media, 50% PBS supplemented with 1% BSA. Under a fluorescent stereoscope, individual fluorescent cells were manually aspirated with a glass pipette into PBS with 0.5% BSA, and then serially transferred until isolated without any additional cellular debris present. Ten cells per sample were aspirated together, transferred to a mini-well containing 3 µL lysis solution (0.2% Triton X-100 in water with 2 U/µL RNAse Inhibitor), lysed by pipetting up and down several times, transferred to a microtube, and stored at –80°C. For the picked cells, 2.3 µL of lysis solution was used as input for library preparation.

### RNA-seq library preparation

RNA-seq libraries were prepared from the picked cells following the Smart-Seq2 protocol for full-length transcriptomes (**Picelli et al., 2014**). To minimize batch effects, primers, enzymes, and buffers were all used from the same lots for all libraries. Libraries were multiplexed, pooled, and purified using AMPure XP beads, quality was checked on an Agilent TapeStation, and libraries were sequenced as 51 bp single end reads on a HiSeq4000 at the UCSF Center for Advanced Technology.

### RNA-seq data analysis

Reads were demultiplexed with CASAVA (Illumina) and read quality was assessed using FastQC (https://www.bioinformatics.babraham.ac.uk/) and MultiQC (**Ewels et al., 2016**). Reads containing adapters were removed using Cutadapt version 2.4 (**Martin, 2011**) and reads were mapped to the *Drosophila melanogaster* transcriptome, FlyBase genome release 6.29, using Kallisto version 0.46.0 (**Bray et al., 2016**) with default parameters. AA samples were removed from further analysis for poor quality, including low read depth (<500,000 reads) and low mapping rates (<80%). Raw sequencing reads and gene expression estimates are available in the NCBI SRA and in the GEO under accession number GSE284380.

## Statistical analysis

For each experimental assay, control populations were sampled to estimate appropriate sample numbers to allow detection of ~33% differences in means with 80% power over a 95% confidence interval. Details of statistical tests including treatment groups, sample numbers (which correspond

to independent biological replicates), statistical tests, p-values, and q-values are provided in *Supplementary file 1*.

## Acknowledgements

This work was supported by grants from the National Institutes of Health to JZP (NINDS R01 NS076614; NINDS R21NS125795), CRW (5F31NS106775), and the MBL (R25NS063307); a grant from the National Science Foundation to SSM (NSF GRFP DGE1752814); funding from the Leading Initiative for Excellent Young Researchers (LEADER) from MEXT, JSPS (KAKENHI 22K06309), and AMED-PRIME (JP22gm6510011) to KI; a grant from the Weill Neurohub to JZP and DMB; a grant from the Scan Design Foundation, a JSPS long-term fellowship and startup funds from UW (JZP); MEXT Grants-in-Aid for Scientific Research (KAKENHI 16H06456), JSPS (KAKENHI 16H02504), WPI-IRCN, AMED-CREST (JP22gm310010), and JST-CREST to KE; and a fellowship from the Grass Foundation (CEE). DMB is an HHMI investigator. Fly Stocks obtained from the Bloomington *Drosophila* Stock Center (NIH P40OD018537) were used in this study. We thank Jessica Huang, Jordan Martel, and David Shen for assistance with video tracking, and Peter Soba for helpful discussions.

## Additional information

### Competing interests

Diana M Bautista: is on the scientific advisory board of Escient Pharmaceuticals. The other authors declare that no competing interests exist.

### Funding

| Funder | Grant reference number | Author |
|---|---|---|
| National Institutes of Health | R01NS076614 | Diana M Bautista<br>Jay Z Parrish |
| National Institutes of Health | R21NS125795 | Jay Z Parrish |
| National Institutes of Health | 5F31NS106775 | Claire R Williams |
| National Institutes of Health | R25NS063307 | Diana M Bautista |
| National Science Foundation Graduate Research Fellowship Program | DGE1752814 | Sonali S Mali |
| Japan Society for the Promotion of Science | KAKENHI 22K06309 | Kenichi Ishii |
| Japan Agency for Medical Research and Development | JP22gm6510011 | Kenichi Ishii |
| Japan Society for the Promotion of Science | KAKENHI 16H02504 | Kazuo Emoto |
| Japan Agency for Medical Research and Development | JP22gm310010 | Kazuo Emoto |
| Ministry of Education, Culture, Sports, Science and Technology | KAKENHI 16H06456 | Kazuo Emoto |
| Weill Neurohub | | Diana M Bautista<br>Jay Z Parrish |
| Scan Design Foundation | | Jay Z Parrish |

| Funder | Grant reference number | Author |
|--------|------------------------|--------|

The funders had no role in study design, data collection and interpretation, or the decision to submit the work for publication.

## Author contributions

Jiro Yoshino, Conceptualization, Data curation, Formal analysis, Investigation, Visualization, Methodology, Writing – original draft, Writing – review and editing; Sonali S Mali, Data curation, Formal analysis, Investigation, Visualization, Methodology, Writing – review and editing; Claire R Williams, Data curation, Formal analysis, Investigation, Visualization, Methodology; Takeshi Morita, Investigation, Methodology; Chloe E Emerson, Christopher J Arp, Sophie E Miller, Lydia Thé, Chikayo Hemmi, Mana Motoyoshi, Investigation; Chang Yin, Data curation, Formal analysis, Visualization; Kenichi Ishii, Kazuo Emoto, Supervision, Writing – review and editing; Diana M Bautista, Conceptualization, Supervision, Funding acquisition, Writing – original draft, Writing – review and editing; Jay Z Parrish, Conceptualization, Data curation, Formal analysis, Supervision, Funding acquisition, Investigation, Visualization, Methodology, Writing – original draft, Writing – review and editing

## Author ORCIDs

Jiro Yoshino (iD) https://orcid.org/0000-0001-9761-4882
Sonali S Mali (iD) https://orcid.org/0000-0003-0737-813X
Claire R Williams (iD) https://orcid.org/0000-0001-5467-149X
Takeshi Morita (iD) https://orcid.org/0000-0002-8570-6744
Chloe E Emerson (iD) https://orcid.org/0000-0003-1188-0501
Christopher J Arp (iD) https://orcid.org/0000-0002-5059-6178
Sophie E Miller (iD) https://orcid.org/0000-0001-6805-7036
Lydia Thé (iD) https://orcid.org/0009-0008-0297-776X
Chikayo Hemmi (iD) https://orcid.org/0009-0002-3815-8164
Mana Motoyoshi (iD) https://orcid.org/0009-0005-4759-0065
Kenichi Ishii (iD) https://orcid.org/0000-0002-8834-5729
Kazuo Emoto (iD) https://orcid.org/0000-0003-4194-801X
Diana M Bautista (iD) https://orcid.org/0000-0002-6809-8951
Jay Z Parrish (iD) https://orcid.org/0000-0002-0656-9148

Reviewer #1 (Public review): https://doi.org/10.7554/eLife.95379.3.sa1
Reviewer #2 (Public review): https://doi.org/10.7554/eLife.95379.3.sa2
Author response https://doi.org/10.7554/eLife.95379.3.sa3

---

# Additional files

## Supplementary files

Supplementary file 1. Details of statistical analyses performed for this study, including comparison groups, statistical tests, and results.

Supplementary file 2. Larval behaviors scored in this study, including a description and scoring criteria for each behavior.

Supplementary file 3. Recipes for solutions used in imaging and physiology experiments.

Source data 1. Raw data for each experiment sorted by figure panel.

MDAR checklist

## Data availability

Raw sequencing reads and gene expression estimates are available in the NCBI Sequence Read Archive (SRA) and in the Gene Expression Omnibus (GEO) under accession number GSE284380. Raw data used for analyses in this study is presented in the supplementary materials as Source Data and details of statistical analyses are presented in Supplementary File 1.

The following dataset was generated:

| Author(s) | Year | Dataset title | Dataset URL | Database and Identifier |
|---|---|---|---|---|
| Yoshino J, Mali SS, Williams CR, Morita T, Emerson CE, Arp CJ, Miller SE, Yin C | 2024 | *Drosophila* epidermal cells are intrinsically mechanosensitive and modulate nociceptive behavioral outputs | https://www.ncbi.nlm.nih.gov/geo/query/acc.cgi?acc=GSE284380 | NCBI Gene Expression Omnibus, GSE284380 |

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

# Appendix 1

**Appendix 1—key resources table**

| Reagent type (species) or resource | Designation | Source or reference | Identifiers | Additional information |
|---|---|---|---|---|
| Gene (*Drosophila melanogaster*) | Stim | GenBank | FLYB:FBgn 0045073 | |
| Gene (*D. melanogaster*) | Orai | GenBank | FLYB:FBgn 0041585 | |
| Genetic reagent (*D. melanogaster*) | A58-GAL4 | Maintained in the Parrish Lab | Flybase_FBal0181674 | GAL4 driver (epidermis) |
| Genetic reagent (*D. melanogaster*) | Desat1-GAL4 | Bloomington Drosophila Stock Center | BDSC_65405 | GAL4 driver (oenocytes) |
| Genetic reagent (*D. melanogaster*) | GMR15F01-GAL4 | Bloomington *Drosophila* Stock Center | BDSC_45071 | GAL4 driver (trachea) |
| Genetic reagent (*D. melanogaster*) | GMR23C11-GAL4 | Bloomington *Drosophila* Stock Center | BDSC_45123 | GAL4 driver (epidermis) |
| Genetic reagent (*D. melanogaster*) | GMR27H06-GAL4 | Bloomington *Drosophila* Stock Center | BDSC_49440 | GAL4 driver (C4da neurons) |
| Genetic reagent (*D. melanogaster*) | GMR32E11-GAL4 | Bloomington *Drosophila* Stock Center | BDSC_48109 | GAL4 driver (epidermis) |
| Genetic reagent (*D. melanogaster*) | GMR38F11-GAL4 | Bloomington *Drosophila* Stock Center | BDSC_50014 | GAL4 driver (epidermis) |
| Genetic reagent (*D. melanogaster*) | GMR51F01-GAL4 | Bloomington *Drosophila* Stock Center | BDSC_38787 | GAL4 driver (epidermis) |
| Genetic reagent (*D. melanogaster*) | GMR51F10-GAL4 | Bloomington *Drosophila* Stock Center | BDSC_38793 | GAL4 driver (epidermis) |
| Genetic reagent (*D. melanogaster*) | GMR61D08-GAL4 | Bloomington *Drosophila* Stock Center | BDSC_39272 | GAL4 driver (Cho neurons) |
| Genetic reagent (*D. melanogaster*) | GMR77B09-GAL4 | Bloomington *Drosophila* Stock Center | Flybase_FBti0138381 | GAL4 driver (epidermis) |
| Genetic reagent (*D. melanogaster*) | hh-GAL4 | Bloomington *Drosophila* Stock Center | DGGR_118117 | GAL4 driver (epidermis) |
| Genetic reagent (*D. melanogaster*) | Hml-GAL4 | Bloomington *Drosophila* Stock Center | BDSC_6395 | GAL4 driver (hemocytes) |
| Genetic reagent (*D. melanogaster*) | MHC-GAL4 | Bloomington *Drosophila* Stock Center | BDSC_55132 | GAL4 driver (muscle) |
| Genetic reagent (*D. melanogaster*) | NompC-GAL4 | Bloomington *Drosophila* Stock Center | BDSC_36361 | GAL4 driver (C3da neurons) |

*Appendix 1 Continued on next page*

*Appendix 1 Continued*

| Reagent type (species) or resource | Designation | Source or reference | Identifiers | Additional information |
|---|---|---|---|---|
| Genetic reagent (*D. melanogaster*) | Piezo-GAL4 | Bloomington *Drosophila* Stock Center | BDSC_78335 | GAL4 driver (piezo-expressing cells) |
| Genetic reagent (*D. melanogaster*) | ppk-GAL4 | Bloomington *Drosophila* Stock Center | BDSC_32079 | GAL4 driver (C4da neurons) |
| Genetic reagent (*D. melanogaster*) | 21-7-GAL4 | Maintained in the Parrish Lab | Flybase_FBti0131369 | GAL4 driver (MD neurons) |
| Genetic reagent (*D. melanogaster*) | repo-GAL4 | Bloomington *Drosophila* Stock Center | BDSC_7415 | GAL4 driver (glia) |
| Genetic reagent (*D. melanogaster*) | sr-GAL4 | Bloomington *Drosophila* Stock Center | BDSC_26663 | GAL4 driver (apodemes) |
| Genetic reagent (*D. melanogaster*) | ppk-LexA | Maintained in the Parrish Lab | Flybase_FBtp0125814 | LEXA driver (C4da neurons) |
| Genetic reagent (*D. melanogaster*) | NompC-LexA | Bloomington *Drosophila* Stock Center | BDSC_52241 | LEXA driver (C3da neurons) |
| Genetic reagent (*D. melanogaster*) | elav-GAL80 | Maintained in the Parrish Lab | Flybase_FBtp0079702 | GAL80 (pan-neuronal) |
| Genetic reagent (*D. melanogaster*) | tsh-GAL80 | Bloomington *Drosophila* Stock Center | Flybase_FBti0114123 | GAL80 (VNC) |
| Genetic reagent (*D. melanogaster*) | ush-GAL4 | Bloomington *Drosophila* Stock Center | BDSC_36524 | GAL4 driver (epidermis) |
| Genetic reagent (*D. melanogaster*) | Nrg167GFP | Bloomington *Drosophila* Stock Center | BDSC_6844 | Reporter (epidermal cell junctions) |
| Genetic reagent (*D. melanogaster*) | UAS-tdTomato | Bloomington *Drosophila* Stock Center | BDSC_36328 | Reporter (RFP) |
| Genetic reagent (*D. melanogaster*) | UAS-RedStinger | Bloomington *Drosophila* Stock Center | BDSC_8546 | Reporter (NLS-RFP) |
| Genetic reagent (*D. melanogaster*) | UAS0GCaMP6s | Bloomington *Drosophila* Stock Center | BDSC_42749 | Reporter (calcium indicator) |
| Genetic reagent (*D. melanogaster*) | UAS-GCaMP6s | Bloomington *Drosophila* Stock Center | BDSC_42746 | Reporter (calcium indicator) |
| Genetic reagent (*D. melanogaster*) | AOP-GCaMP6s | Bloomington *Drosophila* Stock Center | BDSC_44273 | Reporter (calcium indicator) |
| Genetic reagent (*D. melanogaster*) | AOP-TNT | Maintained in the Parrish Lab | Flybase_FBtp0144631 | Neuronal silencing (tetanus toxin) |
| Genetic reagent (*D. melanogaster*) | UAS-shi-ts | Bloomington *Drosophila* Stock Center | BDSC_44222 | Inducible inhibition of endocytosis |

*Appendix 1 Continued on next page*

*Appendix 1 Continued*

| Reagent type (species) or resource | Designation | Source or reference | Identifiers | Additional information |
|---|---|---|---|---|
| Genetic reagent (*D. melanogaster*) | UAS-CsChrimson | Bloomington *Drosophila* Stock Center | BDSC_55135 | Inducible cation channel |
| Genetic reagent (*D. melanogaster*) | UAS-CsChrimson | Bloomington *Drosophila* Stock Center | BDSC_55136 | Inducible cation channel |
| Genetic reagent (*D. melanogaster*) | UAS-TRPA1 | Bloomington *Drosophila* Stock Center | BDSC_26263 | Inducible cation channel |
| Genetic reagent (*D. melanogaster*) | UAS-GtACR | Bloomington *Drosophila* Stock Center | BDSC_92983 | Inducible anion channel |
| Genetic reagent (*D. melanogaster*) | UAS-luciferase RNAi | Bloomington *Drosophila* Stock Center | BDSC_31603 | RNAi transgene |
| Genetic reagent (*D. melanogaster*) | UAS-RFP-RNAi | Bloomington *Drosophila* Stock Center | BDSC_67852 | RNAi transgene |
| Genetic reagent (*D. melanogaster*) | UAS-fwe-RNAi | Bloomington *Drosophila* Stock Center | BDSC_27323 | RNAi transgene |
| Genetic reagent (*D. melanogaster*) | UAS-Ih-RNAi | Bloomington *Drosophila* Stock Center | BDSC_29574 | RNAi transgene |
| Genetic reagent (*D. melanogaster*) | UAS-Ih-RNAi | Bloomington *Drosophila* Stock Center | BDSC_58089 | RNAi transgene |
| Genetic reagent (*D. melanogaster*) | UAS-Irk1-RNAi | Bloomington *Drosophila* Stock Center | BDSC_42644 | RNAi transgene |
| Genetic reagent (*D. melanogaster*) | UAS-KCNQ-RNAi | Bloomington *Drosophila* Stock Center | BDSC_80446 | RNAi transgene |
| Genetic reagent (*D. melanogaster*) | UAS-MCU-RNAi | Bloomington *Drosophila* Stock Center | BDSC_67897 | RNAi transgene |
| Genetic reagent (*D. melanogaster*) | UAS-orai-RNAi | Bloomington *Drosophila* Stock Center | BDSC_53333 | RNAi transgene |
| Genetic reagent (*D. melanogaster*) | UAS-orai-RNAi | Vienna *Drosophila* Resource Center | VDRC_12221 | RNAi transgene |
| Genetic reagent (*D. melanogaster*) | UAS-pain-RNAi | Bloomington *Drosophila* Stock Center | BDSC_51835 | RNAi transgene |
| Genetic reagent (*D. melanogaster*) | UAS-piezo-RNAi | Vienna *Drosophila* Resource Center | FlyBase_FBst0457216 | RNAi transgene |
| Genetic reagent (*D. melanogaster*) | UAS-piezo-RNAi | Kyoto *Drosophila* Stock Center | Flybase_FBtp0071516 | RNAi transgene |
| Genetic reagent (*D. melanogaster*) | UAS-pyx-RNAi | Bloomington *Drosophila* Stock Center | BDSC_31297 | RNAi transgene |

*Appendix 1 Continued on next page*

*Appendix 1 Continued*

| Reagent type (species) or resource | Designation | Source or reference | Identifiers | Additional information |
|---|---|---|---|---|
| Genetic reagent (*D. melanogaster*) | UAS-stim-RNAi | Bloomington *Drosophila* Stock Center | BDSC_41759 | RNAi transgene |
| Genetic reagent (*D. melanogaster*) | UAS-stim-RNAi | Vienna *Drosophila* Resource Center | FlyBase_FBst0478081 | RNAi transgene |
| Genetic reagent (*D. melanogaster*) | UAS-subdued-RNAi | Vienna *Drosophila* Resource Center | FlyBase_FBst0480747 | RNAi transgene |
| Genetic reagent (*D. melanogaster*) | UAS-Task6-RNAi | Bloomington *Drosophila* Stock Center | BDSC_28016 | RNAi transgene |
| Genetic reagent (*D. melanogaster*) | UAS-Task6-RNAi | Vienna *Drosophila* Resource Center | FlyBase_FBst0471360 | RNAi transgene |
| Genetic reagent (*D. melanogaster*) | UAS-Tmem63-RNAi | Vienna *Drosophila* Resource Center | FlyBase_FBst0470667 | RNAi transgene |
| Genetic reagent (*D. melanogaster*) | UAS-TMCO1-RNAi | Bloomington *Drosophila* Stock Center | BDSC_42896 | RNAi transgene |
| Genetic reagent (*D. melanogaster*) | UAS-TMCO1-RNAi | Bloomington *Drosophila* Stock Center | BDSC_55909 | RNAi transgene |
| Genetic reagent (*D. melanogaster*) | UAS-TrpA1-RNAi | Bloomington *Drosophila* Stock Center | BDSC_66905 | RNAi transgene |
| Genetic reagent (*D. melanogaster*) | UAS-Trpm-RNAi | Bloomington *Drosophila* Stock Center | BDSC_31291 | RNAi transgene |
| Genetic reagent (*D. melanogaster*) | UAS-Trpml-RNAi | Bloomington *Drosophila* Stock Center | BDSC_31294 | RNAi transgene |
| Genetic reagent (*D. melanogaster*) | UAS-wtrw-RNAi | Bloomington *Drosophila* Stock Center | BDSC_51563 | RNAi transgene |
| Cell line (*Homo sapiens*) | Immortalized Human Keratinocytes (HaCaT cells) | Cytion | Cat. #: 300493 | |
| Peptide, recombinant protein | Collagenase type I | Thermo Fisher | Cat. #: 17-100-017 | |
| Chemical compound, drug | Fura-2AM | Thermo Fisher | Cat. #: F1221 | |
| Chemical compound, drug | Pluronic F-127 | Thermo Fisher | Cat. #: P3000MP | |
| Chemical compound, drug | All-trans retinal | Millipore Sigma | Cat. #: R2500 | |
| Chemical compound, drug | Ionomycin | Thermo Fisher | Cat. #: I24222 | |
| Chemical compound, drug | Lanthanum chloride | Millipore Sigma | Cat. #: 211605 | 500 nM |
| Software, algorithm | SPSS | SPSS | RRID:SCR_002865 | |
| Software, algorithm | MATLAB | Mathworks | RRID:SCR_001622 | |

*Appendix 1 Continued on next page*

*Appendix 1 Continued*

| Reagent type (species) or resource | Designation | Source or reference | Identifiers | Additional information |
|---|---|---|---|---|
| Software, algorithm | ImageJ | ImageJ | RRID:SCR_003070 | |
| Software, algorithm | CASAVA | Illumina, http://www.illumina.com/software/genome_analyzer_software.ilmn | | |
| Software, algorithm | FasQC | Babraham bioinformatics, https://www.bioinformatics.babraham.ac.uk/ | | |
| Software, algorithm | Cutadapt v2.4 | *Martin, 2011* | | |
| Software, algorithm | Kallisto version 0.46.0 | *Bray et al., 2016* | | |

