## [Editor Report · eLife Assessment]

This is **important** work and provides a significant advance in our understanding of mechanosensation in the epidermis. The evidence presented is **convincing** and, barring a few minor weaknesses, strongly implicates activation of epidermal cells and store-operated calcium entry in the activation of nociceptive neurons innervating that tissue. This work will be of broad interest to neurobiologists, epithelial cell biologists, and mechanobiologists.

---

## [Referee Report · Reviewer #1 (Public review)]

Summary:

In this meticulously conducted study, the authors show that *Drosophila* epidermal cells can modulate escape responses to noxious mechanical stimuli. First, they show that activation of epidermal cells evokes many types of behaviors including escape responses. Subsequently, they demonstrate that most somatosensory neurons are activated by activation of epidermal cells, and that this activation has a prolonged effect on escape behavior. In vivo analyses indicate that epidermal cells are mechanosensitive and require stored-operated calcium channel Orai. Altogether, the authors conclude that epidermal cells are essential for nociceptive sensitivity and sensitization, serving as primary sensory noxious stimuli.

Strengths:

The manuscript is clearly written. The experiments are logical and complementary. They support the authors' main claim that epidermal cells are mechanosensitive and that epidermal mechanically evoked calcium responses require the stored-operated calcium channel Orai. Epidermal cells activate nociceptive sensory neurons as well as other somatosensory neurons in *Drosophila* larvae, and thereby prolong escape rolling evoked by mechanical noxious stimulation.

Weaknesses:

In several places the text is unclear. For example, core details are missing in the protocols, including the level of LED intensity used, which are necessary for other researchers to reproduce the experiments. Secondly, the rationales are missing for some experiments (for experiments X, Y, and Z). It would be helpful to clarify for your readers why the experiments (for example Figure 3S2) were performed. Finally, for most experiments, the epidermal cells are activated for 60 s, which is long when considering that nocifensive rolling occurs on a timescale of milliseconds. It would be informative to know the shortest duration of epidermal cell activation that is sufficient for observing the behavioral phenotype (prolongation of escape behavior) and activation of sensory neurons.

---

## [Referee Report · Reviewer #2 (Public review)]

Summary:

The authors provide compelling evidence that stimulation of epidermal cells in *Drosophila* larvae results in the stimulation of sensory neurons that evoke a variety of behavioral responses. Further, the authors demonstrate that epidermal cells are inherently mechanoresponsive and implicate a role for store-operated calcium entry (mediated by Stim and Orai) in the communication to sensory neurons.

Strengths:

The study represents a significant advance in our understanding of mechanosensation. Multiple strengths are noted. First, the genetic analyses presented in the paper are thorough with appropriate consideration to potential confounds. Second, behavioral studies are complemented by sophisticated optogenetics and imaging studies. Third, identification of roles for store-operated calcium entry is intriguing. Lastly, conservation of these pathways in vertebrates raise the possibility that the described axis is also functional in vertebrates.

Weaknesses:

The study has a few conceptual weaknesses that are arguably minor. The involvement of store-operated calcium entry implicates ER calcium store release. Whether mechanical stimulation evokes ER calcium release in epidermal cells and how this might come about (e.g., which ER calcium channels, roles for calcium-induced calcium release etc.) remains unaddressed. On a related note, the kinetics of store-operated calcium entry is very distinct from that required for SV release. The link between SOC and epidermal cells-neuron transmission is not reconciled. Finally, it is not clear how optogenetic stimulation of epidermal cells results in the activation of SOC.

Revised manuscript:

The authors have adequately addressed my original concerns.

---

## [Author Response]

The following is the authors’ response to the original reviews

**Reviewer #1 (Public Review):**
Summary.In this meticulously conducted study, the authors show that *Drosophila* epidermal cells can modulate escape responses to noxious mechanical stimuli. First, they show that activation of epidermal cells evokes many types of behaviors including escape responses. Subsequently, they demonstrate that most somatosensory neurons are activated by activation of epidermal cells, and that this activation has a prolonged effect on escape behavior. In vivo analyses indicate that epidermal cells are mechanosensitive and require stored-operated calcium channel Orai. Altogether, the authors conclude that epidermal cells are essential for nociceptive sensitivity and sensitization, serving as primary sensory noxious stimuli.Strengths.The manuscript is clearly written. The experiments are logical and complementary. They support the authors' main claim that epidermal cells are mechanosensitive and that epidermal mechanically evoked calcium responses require the stored-operated calcium channel Orai. Epidermal cells activate nociceptive sensory neurons as well as other somatosensory neurons in *Drosophila* larvae, and thereby prolong escape rolling evoked by mechanical noxious stimulation.Weaknesses.Core details are missing in the protocols, including the level of LED intensity used, which are necessary for other researchers to reproduce the experiments. For most experiments, the epidermal cells are activated for 60 s, which is long when considering that nocifensive rolling occurs on a timescale of milliseconds. It would be informative to know the shortest duration of epidermal cell activation that is sufficient for observing the behavioral phenotype (prolongation of escape behavior) and activation of sensory neurons.

(1) We agree with the reviewer that the LED intensity is an important detail of the experimental paradigm. We updated the methods to include intensity measurements for the stimuli used throughout the manuscript.

(2) The Reviewer asks about the shortest duration of epidermal cell activation sufficient for observing the behavior phenotype. We note in the manuscript that behavioral responses to optogenetic epidermal stimulation are apparent within 2 seconds of stimulus (see Figure 2F); this is consistent with our calcium imaging data in which C4da response reaches its maximum within 2-3 sec of stimulation.

**Reviewer #1 (Recommendations):**
(1) The epidermal cells in this study are activated for 60 s. In the real world, the nociceptive stimulation (a poke, such as penetration by the ovipositor of a parasitic wasp) that evokes escape rolling is short. Does optogenetic activation of 1 s or less still evoke rolling? For example, it is unclear in Figure 4K how long the epidermal cells need to be activated before the poke stimulus prolongs rolling. Is it possible to test behavior and GCaMP activity in sensory neurons when epidermal cells are briefly (1 second) activated?

As described above, behavioral responses to optogenetic epidermal stimulation are apparent within 2 seconds of stimulus (see Figure 2F); this is consistent with our calcium imaging data in which C4da response reaches its maximum within 2-3 sec of stimulation. The kinetics are consistent with a role for epidermal cells in modulating neuronal responses to nocifensive stimuli, and similar to the response kinetics observed in mammalian epidermal cells that modulate neuronal touch and pain responses (Maksimovic et al., 2014; Woo et al., 2014; Mikesell et al., 2022).

(2) The protocol for optogenetic screening states that the authors used a 488-nm LED. Why was a 488-nm LED used instead of the 610-nm LED for Chrimson activation? No information (except figure 4K) about the light intensity is provided in the figure legend or the protocol section. Please state the LED intensity used for all optogenetic experiments (GCaMP imaging, behavioral experiments, etc.).

We used 488 nm light for the initial screen for technical reasons. The screen was conducted by students at the MBL Neurobiology course (hence the affiliation; student authors are included in the manuscript), and the only LED available to us at that time delivered insufficient illumination at longer wavelenths to be useful. We chose to include the student’s data because (1) we found that the 488 nm light alone did not induce rolling in our setup, (2) we repeated and extended the studies with the epidermal drivers using a higher resolution imaging platform and longer wavelength stimulation (all studies other than Fig. 1), and (3) we observed qualitatively similar results when we repeated stimulation with all drivers using 561 nm light.

We agree that the LED intensity is an important detail of the experimental paradigm. We updated the methods to include intensity measurements for the stimuli used throughout the manuscript. We also include the intensities here:

- 30 μW/mm^2 for calcium imaging experiments Fig 3B-E, Fig 4A, Fig 3S1A-D, Fig 4S1A

- 300 μW/mm^2 for behavior studies in Fig 2B-E, Fig 1S6, Fig 2S1, Fig 3E-F, Fig 3S2A-C

- 25 μW/mm^2 for behavior studies in Fig 4E-J

- 1.16 μW/mm^2 for behavior studies in Fig 4K

(3) Lines 150 - 152: Although the authors refer to "a stereotyped behavior sequence" in Fig 2D, there are no data supporting this claim in Fig 2. Rather, the data appear to represent proportions of different types of behavior at each time point, rather than behavior sequences. If the authors wish to claim that the data show stereotyped behavior sequences, they should analyze the data using a different method (e.g., Markov models).

We agree that in the absence of additional analysis we should avoid commenting on stereotypy of behavior sequences; we therefore adjusted the text to reflect the tendency of nociceptive behaviors to precede non-nociceptive behaviors. The raster plots shown in Supplemental Fig. 2A illustrate this point: in larvae exhibiting nociceptive behaviors, these behaviors appear first, followed by backing and frequently freezing. As one quantitative readout of this sequence we show that the latency of rolling (nociceptive) is shorter compared with backing or freezing (non-nociceptive) (Fig. 2F, Fig. S2G).

(4) Figure 3A-E: a cursory glance at the data suggests that the most responsive sensory neurons are C1da, with all sensory neurons activated. However, at the behavioral level, only some sensory neurons are activated. If all sensory were activated by Chrimson, what behavioral phenotypes would the authors expect to see? Would it be the same as epidermal activation?

The Reviewer raises an interesting question, but we intentionally avoid comparing the response properties among sensory neurons because of differences in driver strength. Likewise, extrapolating “activation” at the behavioral level is exceedingly difficult if/when multiple sensory neurons are simultaneously activated. In response to the Reviewer’s specific question, when all da neurons are activated simultaneously, larvae largely exhibited hunching rather than rolling (Hwang et al., 2007). We find that epidermal stimulation rarely elicits hunching; instead, epidermal stimulation generally triggers nocifensive behaviors followed by non-nocifensive behaviors such as backing and freezing, suggesting an order or priority in neurons activated by epidermal cells (or different response times). Defining the mechanisms by which epidermal cells communicate with different types of sensory neurons is therefore a top priority for future studies.

(5) Figure 3S2; The behavior phenotypes between Fig. 3E, F and Fig 3S2 seems a slightly different. I suggest adding some comments in different behavior phenotype depending on the different GAL4. Specifically, is there increased freezing in some genotypes (e.g., ppk-LexA or NompC-lexA)? Can you show this without TNT data? Is this a background effect or specific GAL4 phenotype?

We currently do not have the driver-only control for this experiment, but our effector-only control experiment (see Fig. 3S2A) suggests that larvae carrying the AOP-TNT insertion exhibit enhanced nociceptive behavioral responses. This point is addressed in our manuscript by the following (copied from the figure legend):

“We note that although baseline rolling probability is elevated in all genetic backgrounds containing the AOP-LexA-TnT insertion, silencing C4da and C3da neurons significantly attenuates responses to epidermal stimulation.”

(6) Calcium-free solution is used in Figure 3. Why do the authors still observe calcium influx? Does this mean that internal calcium stores are released? If so, does the calcium influx represent an action potential? How do the authors focus their LED stimulation to activate epidermal cells and avoid activation of the imaging laser?

The specimens were imaged in calcium-free solution to minimize movement artifacts. However, the CNS is wrapped by glial cells and over short timescales such as those used for the imaging we speculate that extracellular calcium persists in the CNS.

(7) It is unclear when animals begin to crawl after the epidermal cells are mechanically stimulated. How do the authors distinguish between peristaltic crawling and a poke by Orai receptors? Although the in vitro experiments beautifully show radial tensions, it is unclear to what extent A-P axis tension (peristaltic crawling) and radial tension (poke) differ. It might be helpful to explain in the discussion section how epidermal cells are selectively activated.

The Reviewer raises an interesting question about the types and thresholds of forces required to elicit epidermal responses. We cannot eliminate the possibility that peristaltic crawling (or crawling through a 3D substrate) stimulates epidermal cells to a certain degree. Indeed, our results demonstrate a dose-dependent response of *Drosophila* epidermal cells and human keratinocytes to radial stretch. However, we do not have any information about selectivity in response to different stimuli, though we agree that this is an intriguing avenue for future studies. For example, we don't know whether stretch-responsive cells are more or less responsive to poke. But, a salient feature of our studies is the recruitment of greater numbers of responders with increasing stimulus intensity, therefore we added the following statement to the discussion to clarify our model:

“Finally, we find that epidermal cells exhibit a dose-dependent response to radial stretch; we therefore anticipate that the output of epidermal cells is likewise dependent on the stimulus intensity. Hence, rather than a fixed threshold beyond which epidermal cells are selectively activated, we hypothesize that increasing stimulus intensities drive increasing signal outputs to neurons.”

(8) Some Protocols are missing. For example, in Figure 4, many stimulus combinations were used to test behavior. How were stimuli of different modalities applied to the animals? Further details need to be provided in the protocols.

We thank the Reviewer for identifying this oversight. The methods section of our original submission detailed most of the stimulus combinations but omitted the opto + mechano combination (4F). We updated our methods to correct these omissions.

(9) It might be helpful if the authors could provide a sample video for each behavior to clarify how they were each defined.

Our manuscript includes a table with a detailed description of the behaviors (Table S2), and we added two annotated videos that show representative behavioral responses to optogenetic nociceptor or epidermis stimulation.

(10) A supplementary summary table of genotypes might be helpful for the reader.

Experimental genotypes are provided in the figure legends, and a detailed list of all alleles used in the study as well as their source is provided in supplemental table S1.

**Reviewer #2 (Public Review):**
Summary.The authors provide compelling evidence that stimulation of epidermal cells in *Drosophila* larvae results in the stimulation of sensory neurons that evoke a variety of behavioral responses. Further, the authors demonstrate that epidermal cells are inherently mechanoresponsive and implicate a role for store-operated calcium entry (mediated by Stim and Orai) in the communication to sensory neurons.Strengths.The study represents a significant advance in our understanding of mechanosensation. Multiple strengths are noted. First, the genetic analyses presented in the paper are thorough with appropriate consideration to potential confounds. Second, behavioral studies are complemented by sophisticated optogenetics and imaging studies. Third, identification of roles for store-operated calcium entry is intriguing. Lastly, conservation of these pathways in vertebrates raise the possibility that the described axis is also functional in vertebrates.Weaknesses.The study has a few conceptual weaknesses that are arguably minor. The involvement of store-operated calcium entry implicates ER calcium store release. Whether mechanical stimulation evokes ER calcium release in epidermal cells and how this might come about (e.g., which ER calcium channels, roles for calcium-induced calcium release etc.) remains unaddressed. On a related note, the kinetics of store-operated calcium entry is very distinct from that required for SV release. The link between SOC and epidermal cells-neuron transmission is not reconciled. Finally, it is not clear how optogenetic stimulation of epidermal cells results in the activation of SOC.(1) The involvement of store-operated calcium entry implicates ER calcium store release. Whether mechanical stimulation evokes ER calcium release in epidermal cells and how this might come about (e.g., which ER calcium channels, roles for calcium-induced calcium release etc.) remains unaddressed.

Our studies suggest that mechanically evoked responses in epidermal cells involve both ER calcium release and store-operated calcium entry. Notably, we show that depletion of ER calcium stores before mechanical stimulation, by treating with thapsigargin, reduces (but does not eliminate) mechanically evoked calcium responses in fly epidermal cells (Fig. 6C-6F). Likewise, fly epidermal cells and human keratinocytes both exhibit mechanically evoked calcium responses in the absence of extracellular calcium (10mM EGTA to chelate all free calcium ions). These data support a model whereby mechanical stimuli trigger calcium release from ER stores and influx. Indeed, several cell types have been shown to display mechanically evoked release of calcium from stores. For example, mechanical stimulation of enteroendocrine cells of the gut epithelium results in both calcium release from ER stores and calcium influx across the plasma membrane (Knutson et al., 2023). Similar to our findings, Knutson et al found that depleting stores decreased mechanically evoked calcium signals by over 70% in these gut epithelial stores. In our revised manuscript we have more clearly emphasized these points.

We agree with the reviewer that deciphering the mechanisms by which mechanical stimuli promote ER calcium release and subsequent store-operated calcium entry is an exciting topic to explore. One potential mechanism is the activation of a mechanosensitive receptor that promotes calcium release from the ER via calcium-induced calcium release or IP3 production, as has been proposed for enteroendocrine cells. A recent paper demonstrated that the ER itself is mechanosensitive and that mechanical stimuli promotes calcium release via the opening of calcium-permeable ion channels in the ER membrane (Song et al., 2024). Determining the relative contributions of store-operated calcium entry and ER calcium release and deciphering their underlying mechanisms will require a thorough investigation of ER calcium channels and receptors, thus we believe this would be beyond the scope of the present manuscript and merits publication on its own. However, we now include this in our discussion as an exciting new direction we aim to pursue.

(2) The kinetics of store-operated calcium entry is very distinct from that required for SV release. The link between SOC and epidermal cells-neuron transmission is not reconciled.

The Reviewer raises an interesting point regarding the mode of epidermal cell-neuronal communication. We demonstrated a requirement for dynamin-dependent vesicle release from epidermal cells in mechanical sensitization. However, the nature of the vesicular pool, the mode and kinetics of release, and the type of neuromodulator released remain to be characterized. Hence, it’s not clear that kinetics of synaptic vesicle release is an appropriate comparison. Our studies do demonstrate that behavioral responses to optogenetic epidermal stimulation are relatively slow – on the order of seconds – which is not incompatible with the kinetics of store-operated calcium entry. Furthermore, the primary functional output we define for epidermal mechanosensory responses, mechanical nociceptive sensitization, is apparent 10 sec following the stimulus and persists for minutes in our behavior assays. Consistent with this model, studies of the mammalian touch dome have shown that touch-sensitive Merkel cells secrete neurotransmitters to modulate neurons and promote sustained action potential firing on a similar timescale. Likewise, mechanically evoked ER calcium-release promotes sustained secretion of serotonin from enterochromaffin cells.

(3) It is not clear how optogenetic stimulation of epidermal cells results in the activation of SOC.

We appreciate the opportunity to clarify our results. We demonstrate that optogenetic epidermal stimulation elicits behavioral responses in larvae and calcium responses in somatosensory neurons, but we do not claim that optogenetic epidermal stimulation elicits SOC. Our optogenetic studies demonstrate the capacity for epidermal stimulation to modulate somatosensory function, but we characterize contributions of SOC only to mechanical stimuli which are more physiologically relevant. However, it is worth noting that CsChrimson is a calcium-permeable channel, suggesting that an increase in intracellular calcium may trigger epidermal-evoked neuronal responses and behaviors during optogenetic stimulation.

References

Hwang, RY, Zhong, L, Xu, Y, Johnson, T, Zhang, F, Deisseroth, K, and Tracey, WD (2007). Nociceptive neurons protect *Drosophila* larvae from parasitoid wasps. Curr Biol 17, 2105–2116.

Knutson, KR, Whiteman, ST, Alcaino, C, Mercado-Perez, A, Finholm, I, Serlin, HK, Bellampalli, SS, Linden, DR, Farrugia, G, and Beyder, A (2023). Intestinal enteroendocrine cells rely on ryanodine and IP3 calcium store receptors for mechanotransduction. J Physiol 601, 287–305.

Maksimovic, S, Nakatani, M, Baba, Y, Nelson, AM, Marshall, KL, Wellnitz, SA, Firozi, P, Woo, S-H, Ranade, S, Patapoutian, A, et al. (2014). Epidermal Merkel cells are mechanosensory cells that tune mammalian touch receptors. Nature 509, 617–621.

Mikesell, AR, Isaeva, O, Moehring, F, Sadler, KE, Menzel, AD, and Stucky, CL (2022). Keratinocyte PIEZO1 modulates cutaneous mechanosensation. Elife 11, e65987.

Song, Y, Zhao, Z, Xu, L, Huang, P, Gao, J, Li, J, Wang, X, Zhou, Y, Wang, J, Zhao, W, et al. (2024). Using an ER-specific optogenetic mechanostimulator to understand the mechanosensitivity of the endoplasmic reticulum. Dev Cell 59, 1396-1409.e5.

Woo, S-H, Ranade, S, Weyer, AD, Dubin, AE, Baba, Y, Qiu, Z, Petrus, M, Miyamoto, T, Reddy, K, Lumpkin, EA, et al. (2014). Piezo2 is required for Merkel-cell mechanotransduction. Nature 509, 622–626.